METHODS

# Bayesian-calibrated global sensitivity analysis for mathematical models using generative AI

Xuyuan Wang*

Department of Mathematical and Statistical Sciences, University of Alberta, Edmonton, Alberta, Canada

* xuyuan@ualberta.ca

## Abstract

We present a generative modeling framework for global sensitivity analysis (GSA) in complex systems characterized by strong and potentially high-dimensional parameter correlations. Traditional variance-based GSA methods rely on the assumption of independent inputs, which rarely holds for Bayesian-calibrated models. While recent extensions using Rosenblatt transformations and Shapley effects theoretically address this limitation, their implementation requires accurate conditional sampling from correlated joint distributions, a task that remains challenging. Existing solutions suffer from restrictive assumptions on input dependence, which limit their applicability to complex data-driven problems. Our method addresses these challenges by reframing sensitivity analysis as a post calibration task on Bayesian posterior distributions, where parameter correlations are learned from data using generative models, eliminating restrictive dependence assumptions and ensuring data relevant sensitivity estimates. We employ autoregressive architectures to implement Rosenblatt transformations and leverage diffusion models to estimate Shapley effects. These methods impose no predefined distributional assumptions and scale efficiently with both data volume and model complexity. We demonstrate the effectiveness of our approach on two representative applications: a COVID-19 transmission model and a cancer immunotherapy model. Results show that our methods effectively captures parameter sensitivities in the presence of parameter correlations, and achieve notable gains in scalability and flexibility over existing methods.

## Author summary

In this research, we introduce a novel approach for conducting global sensitivity analysis in biological models using generative AI. Our method is fully compatible with Bayesian inference, which is widely used for parameter calibration of biological systems. Unlike traditional sensitivity analyses that assume independent parameters or impose simplified dependence structures, our approach performs sensitivity analysis directly on Bayesian-calibrated posterior distributions, where

**Data availability statement:** All data and source code are publicly available. The Wuhan COVID-19 dataset can be found in https://weekly.chinacdc.cn/news/TrackingtheEpidemic.htm, and the patient-specific CAR-T cell treatment data is available in https://ascpt.onlinelibrary.wiley.com/doi/abs/10.1002/cpt.2040. The code used to generate results is provided in https://doi.org/10.5281/zenodo.15685729.

**Funding:** The author(s) received no specific funding for this work.

**Competing interests:** The author has declared that no competing interests exist.

parameter correlations are learned from observational data. As a result, the resulting sensitivity analysis reflects realistic, data relevant parameter sensitivities rather than purely structural sensitivities of an abstract model. The proposed framework is flexible, scalable, and broadly applicable to a wide range of deterministic models calibrated through Bayesian methods. Furthermore, the generative nature of the approach paves the way for future extensions to distributional sensitivity analysis in stochastic or agent-based models, enhancing its potential for modern biological applications.

## Introduction

Mathematical modeling, Bayesian inference, and sensitivity analysis are important tools for conducting complex, data-driven research projects across diverse scientific fields. Real-world problems in disciplines such as health science are frequently analyzed using mathematical models because of their ability to abstract complex system behaviors and describe them through parameterized equations. These models enable researchers to simulate and predict how systems respond to varying conditions, providing valuable insights into their underlying dynamics. Central to the reliability of these predictions is the accurate estimation of model parameters, a task effectively addressed by Bayesian inference [1]. This statistical approach updates prior beliefs about parameter values using observed data, producing a posterior distribution that captures both parameter uncertainty and interdependencies. Based on the parameter estimates, we can then conduct sensitivity analysis to identify which parameters have the greatest influence on model outcomes.

There are various approaches within the broad scope of sensitivity analysis. For example, local sensitivity analysis focuses on assessing the effects of small perturbations applied to individual model parameters, whereas global sensitivity analysis (GSA) examines the impact of varying multiple parameters simultaneously across a specified range. In this paper, we focus on a GSA method, specifically the variance-based approach known as Sobol's method [2]. This method quantifies parameter sensitivity by estimating the portion of the output variance that can be attributed to variations in each individual input parameter, as well as their interactions. The resulting first order (individual) and higher order (interaction) indices provide a comprehensive measure of parameter influence. Sobol's method has gained widespread adoption across various domains due to its robustness and interpretability, as demonstrated in the following studies [3–5].

Despite their widespread utility, estimating Sobol's sensitivity indices on real world mathematical models presents significant challenges, primarily due to the intricate correlations embedded in the Bayesian posterior distribution of model parameters conditioned on observed data. Traditional Sobol's method typically assume that input parameters are independent, whereas real world systems often exhibit complex, high-dimensional dependencies arising from the interaction between model structure and data. These dependencies limit the applicability of the method, as the

corresponding Monte Carlo estimators become biased when the independence assumption is violated, potentially leading to misleading conclusions about parameter sensitivities. To address this issue, several approaches have been proposed. For example, Rosenblatt transformations can be used to map any complex joint distribution into independent uniform variables over the unit hypercube, thereby enabling the use of traditional Sobol's method [6]. While theoretically sound for arbitrarily complex distributions, their practical implementation requires approximating a sequence of conditional density functions, which is highly non-trivial. An alternative approach employs Shapley effects from cooperative game theory to quantify input sensitivities under correlation [7]. Shapley effects naturally accommodate dependencies by averaging each input's marginal contribution across all possible subsets of parameters. However, estimating Shapley effects also relies on conditional sampling from arbitrary subsets of the joint distribution, presenting similar computational challenges. From these examples, it becomes clear that accurate sampling from conditional distributions of model inputs is central to performing GSA. Existing copula based methods have attracted attention due to their practical simplicity [8]. By separating the marginal distributions from the dependence structure, copulas allow very efficient conditional sampling as long as the correlation structure is accurately represented by the copula function. In practice, Gaussian copulas are widely used but impose restrictive assumptions, such as symmetric and linear dependence, which may fail to model complex dependencies among inputs. Additionally, they can lose accuracy in high-dimensional settings due to the curse of dimensionality [9].

Recent advances in generative artificial intelligence offer a promising solution for accurately sampling from complex conditional parameter distributions in global sensitivity analysis. Methods such as autoregressive models, which sequentially model variable dependencies, and diffusion based methods, which learn the joint distribution and generate flexible conditional samples, have demonstrated remarkable capabilities in capturing high-dimensional, non-Gaussian distributions [10]. Inspired by the flexibility and scalability of generative models, we propose reframing GSA as a post calibration task performed directly on Bayesian posterior distributions, in which parameter correlations arise naturally from the interaction between model structure and observational data. By learning joint and conditional parameter distributions from posterior samples using generative models, the proposed framework eliminates restrictive independence or dependence assumptions. Moreover, by incorporating model outputs as additional features of the target joint distribution alongside the inputs, the approach has the potential to decouple sensitivity analysis from repeated forward model evaluations, thereby reducing the computational burden when model evaluations are expensive. As a result, the resulting sensitivity indices reflect realistic, Bayesian-calibrated parameter sensitivities, rather than purely structural sensitivities of an abstract, uncalibrated model.

In this paper, we explore how generative models can be leveraged to conduct GSA for Bayesian-calibrated mathematical models. The Materials section provides background on the mathematical modeling framework, Bayesian data calibration, and Sobol's sensitivity analysis. The Methods section introduces the proposed generative modeling approach for GSA. The Case Studies Results section validates the methodology through numerical experiments, including both synthetic benchmarks and real world applications in the health sciences.

## Materials

### Mathematical Models

Mathematical models are important tools for representing and analyzing complex systems across a wide range of disciplines. These models can be broadly categorized by their mathematical structure and application domains. Deterministic models, such as Ordinary Differential Equations (ODEs) and Partial Differential Equations (PDEs), are widely used in the physical sciences [11] and health sciences [12] to describe systems with spatial and temporal dynamics. Their deterministic nature facilitates data calibration, as they typically yield tractable likelihood functions that are easier to compute and optimize. While this manuscript focuses on ODE based models due to their prevalent use in health science research, the proposed methodology is designed to be applicable to the class of deterministic mathematical models.

A dynamical system model represents a system using a set of ODEs to describe the relationships among variables, defined as

$$\frac{d\boldsymbol{x}}{dt} = \boldsymbol{F}(\boldsymbol{x}, t, \theta),$$
(1)

where $\boldsymbol{x}(t) \in \mathbb{R}^n$ is an $n$-dimensional vector representing the state variables, $t$ is time, and $\theta \in \mathbb{R}^m$ is an $m$-dimensional vector of model parameters (inputs). The function $\boldsymbol{F}: \mathbb{R}^n \times \mathbb{R}^+ \times \mathbb{R}^m \to \mathbb{R}^n$ is a continuously differentiable mapping that defines the system dynamics.

Once the model is properly defined, the model output can be defined based on the state variables and model parameters as

$$y = g(\boldsymbol{x}, \theta),$$
(2)

where $y \in \mathbb{R}$ is the output, and $g$ is a known real-valued function. In the sensitivity analysis literature, the model output is also referred to as the *quantity of interest* (QoI). When observational data are available, we assume that these data correspond directly to the model output. This assumption is made for notational simplicity in the calibration process. In practice, the function $g$ can be defined with considerable flexibility, and multiple outputs can be specified, not all of which need to correspond directly to the observed data.

## Bayesian Inference

The mathematical model defined in Equation (1) remains theoretical unless its parameters, $\theta$, are calibrated to reflect real-world phenomena. To achieve this, the model must be fitted to available data, denoted as

$$D = \{(t_1, y_1), (t_2, y_2), \dots, (t_k, y_k)\}.$$

Bayesian inference combined with Markov Chain Monte Carlo (MCMC) algorithms is commonly used for parameter calibration. We begin by assuming a prior distribution over the model parameters, $\theta \sim p_0(\theta)$, which encodes our prior knowledge. According to Bayes' theorem, the posterior distribution of the parameters given the data $D$ satisfies

$$p(\theta|D) = \frac{L(D|\theta)\, p_0(\theta)}{p(D)} \propto L(D|\theta)\, p_0(\theta),$$
(3)

where $p(\theta|D)$ is the posterior distribution, $L(D|\theta)$ is the likelihood function, and $p(D)$ is the marginal likelihood, which serves as a normalization constant.

The choice of likelihood function $L$ is critical, as it measures the discrepancy between the observed data and the simulated model outputs. In practice, it is often assumed that the dataset $D$ consists of independent realizations of a data-generating random variable, whose mean is modeled by the output $y$. Depending on the characteristics of the data, common choices for the distribution of this random variable include the normal, Poisson, or negative binomial distributions. For a detailed discussion on likelihood selection, we refer the reader to [13].

Since analytical solutions to the posterior distribution are typically intractable, MCMC algorithms are employed to draw samples from the posterior. Given sufficient convergence and a large enough sample size, the resulting posterior sample set $S = \{\theta_i\}_{i=1}^N$ provides a good approximation of the true posterior distribution, along with the corresponding posterior predictive outputs simulated for each sample, denoted as $S_y = \{g(\boldsymbol{x}_i, \theta_i)\}_{i=1}^N$. The sample sets $S$ and $S_y$ encapsulate the information that the data $D$ provides about the statistical properties of the model parameters and their associated outputs. We quantify model uncertainty for both $\theta$ and $y$ using these sample sets by evaluating Bayesian credible intervals. Overall,

Bayesian inference in mathematical modeling quantifies uncertainty in parameters and outputs given the data, providing a foundation for sensitivity analysis.

## Global Sensitivity Analysis

After the model calibration process, we are interested in quantifying the influence of input parameters on model outputs within a plausible range of parameter values. This is the primary objective of GSA. Several methods have been proposed in the literature, including the Derivative based Global Sensitivity Measure (DGSM) [14], Partial Rank Correlation Coefficient (PRCC) [15], and variance based approaches [16]. Among these, variance based methods, particularly those based on Sobol' indices, are widely adopted due to their rigorous statistical foundation and intuitive interpretation. Sobol' indices are derived from the variance decomposition of a square-integrable model output $y = g(\boldsymbol{x}, \boldsymbol{\theta})$ with respect to the input parameters $\boldsymbol{\theta} = (\theta_1, \ldots, \theta_m) \in \mathbb{R}^m$. Assume $\boldsymbol{\theta} \sim U[0, 1]^m$, the total variance of the output can be decomposed as

$$\text{Var}(y) = \sum_{i=1}^{m} V_i + \sum_{i<j} V_{ij} + \sum_{i<j<k} V_{ijk} + \cdots + V_{1,2,\ldots,m},$$

where $V_i = \text{Var}_{\theta_i}(\mathbb{E}_{\theta_{\sim i}}[y \mid \theta_i])$, $V_{ij} = \text{Var}_{\theta_i, \theta_j}(\mathbb{E}_{\theta_{\sim\{i,j\}}}[y \mid \theta_i, \theta_j]) - V_i - V_j$, and higher-order terms capture interactions among parameters. Two commonly used indices derived from this decomposition are the first order Sobol' index ($S_i$) and the total order Sobol' index ($S_{T_i}$). The first order index measures the effect of parameter $\theta_i$ alone, while the total order index captures the contribution of $\theta_i$ including all its interactions with other parameters. These indices are defined as

$$S_i = \frac{\text{Var}_{\theta_i}\left(\mathbb{E}_{\theta_{\sim i}}[y \mid \theta_i]\right)}{\text{Var}(y)}, \quad S_{T_i} = 1 - \frac{\text{Var}_{\theta_{\sim i}}\left(\mathbb{E}_{\theta_i}[y \mid \theta_{\sim i}]\right)}{\text{Var}(y)}, \tag{4}$$

where $\mathbb{E}_{\theta_i}[y \mid \theta_{\sim i}]$ denotes the conditional expectation of $y$ given all parameters except $\theta_i$, and $i \in \{1, 2, \cdots, m\}$.

Estimating these indices (4) requires accurate computation of inner conditional expectations and outer conditional variances, which can be computationally expensive. Several Monte Carlo based methods have been proposed to address this challenge. For example, I.M. Sobol' et al. [17] introduced the well-known two-matrix estimator, which divides the posterior sample set into two independent subsets

$$S_A = \left\{(\theta_i^{(A)}, y_i^{(B)})\right\}_{i=1}^{N/2}, \quad S_B = \left\{(\theta_i^{(B)}, y_i^{(B)})\right\}_{i=1}^{N/2}.$$

To estimate the Sobol' indices for parameter $\theta_j$, the method constructs a hybrid sample by replacing the $\theta_j$ values in $S_A$ with those from $S_B$, resulting in the model output $\boldsymbol{y}^{(AB)}$. The estimators for the first order and total order indices are then given by

$$\widehat{S}_j = \frac{\frac{1}{N}\sum_{i=1}^{N} y_i^{(B)}(y_i^{(AB)} - y_i^{(A)})}{\widehat{V}}, \quad \widehat{S}_{T_j} = \frac{\frac{1}{N}\sum_{i=1}^{N}\left(y_i^{(AB)} - y_i^{(A)}\right)^2}{2\widehat{V}}, \tag{5}$$

for $j = 1, 2, \ldots, m$, where $\widehat{V}$ is the unbiased variance estimator of the model output $y$ computed from the full posterior sample set $S$. Despite its numerical efficiency, this approach has limitations. The swapping process assumes that the input distribution is independent and uniform, an assumption that is rarely satisfied in real-world applications.

Another widely used estimator is the extended Fourier Amplitude Sensitivity Test (eFAST), which estimates Sobol' indices through spectral decomposition [18]. While eFAST can accommodate some non-uniform distributions, it still assumes input independence. Additionally, the sinusoidal search curves must be carefully selected to reflect the true posterior

distribution, which becomes impractical when parameter interdependencies are complex. As a result, the proper treatment of highly correlated model parameters is by no means trivial and needs further investigation.

**Handling Correlated Parameters**

While the complex correlation structure in $\theta$ poses challenges to GSA, several promising approaches have been proposed to address this issue. Thierry A. Mara et al. [6] introduced the use of the Rosenblatt transformation to map the correlated input parameters $\theta$ onto independent uniform variables over $[0, 1]^m$. Specifically, the Rosenblatt transformation associated with the $i$-th natural permutation $RT_i : \mathbb{R}^m \mapsto [0, 1]^m$ maps correlated parameters $\theta = (\theta_1, \theta_2, \ldots, \theta_m)$ to independent variables $\mathbf{Z} = (Z_1, Z_2, \ldots, Z_m)$ as follows

$$
\begin{aligned}
Z_i &= F_i(\theta_i), \\
Z_{i+1} &= F_{i+1}(\theta_{i+1} \mid \theta_i), \\
Z_{i+2} &= F_{i+2}(\theta_{i+2} \mid \theta_i, \theta_{i+1}), \\
&\vdots \\
Z_{i-1} &= F_{i-1}(\theta_{i-1} \mid \theta_i, \ldots, \theta_{i-2}),
\end{aligned}
\tag{6}
$$

where $F_i$ denotes the conditional cumulative distribution function (CDF) for each $i \in \{1, 2, \cdots, m\}$. This transformation enables the application of traditional two-matrix estimator to problems involving correlated inputs. Among the components in $RT_i$ (6), we are particularly interested in $Z_i$ and $Z_{i-1}$, as they represent the full and independent contributions of $\theta_i$ and $\theta_{i-1}$, respectively. Based on this, two types of sensitivity indices are defined: one that captures the effect of a parameter in isolation (i.e., uncorrelated) and one that includes its correlations with others.

$$
\begin{aligned}
S_i^{\text{full}} &= \frac{\text{Var}_{Z_i}\left(\mathbb{E}_{\mathbf{z}_{\sim i}}[y \mid Z_i]\right)}{\text{Var}(y)}, \quad & S_{T_i}^{\text{full}} &= 1 - \frac{\text{Var}_{\mathbf{z}_{\sim i}}\left(\mathbb{E}_{Z_i}[y \mid \mathbf{Z}_{\sim i}]\right)}{\text{Var}(y)}, \\
S_{i-1}^{\text{ind}} &= \frac{\text{Var}_{Z_i}\left(\mathbb{E}_{\mathbf{z}_{\sim i-1}}[y \mid Z_i]\right)}{\text{Var}(y)}, \quad & S_{T_{i-1}}^{\text{ind}} &= 1 - \frac{\text{Var}_{\mathbf{z}_{\sim i-1}}\left(\mathbb{E}_{Z_{i-1}}[y \mid \mathbf{Z}_{\sim i-1}]\right)}{\text{Var}(y)}.
\end{aligned}
\tag{7}
$$

We refer to the indices that account for the full correlated effects of the parameters, namely $S_i^{\text{full}}$ and $S_{T_i}^{\text{full}}$, as the full Sobol' indices. In contrast, the indices $S_{i-1}^{\text{ind}}$ and $S_{T_{i-1}}^{\text{ind}}$, which are based on the independent components derived from the Rosenblatt transformation, are referred to as the independent Sobol' indices.

The Rosenblatt transformation provides a theoretically sound method to handle correlated model inputs by transforming them into an independent space, allowing direct interpretation in sensitivity analysis. Another approach, proposed by E. Song et al. [7], combines the Shapley effect from game theory with Sobol' indices to directly account for conditional dependencies in the input parameters $\theta = (\theta_1, \ldots, \theta_m) \in \mathbb{R}^m$. The Shapley effect estimates the average marginal contribution of a parameter $\theta_i$ to the output variance, conditioned on all possible subsets of the remaining parameters. The Shapley effect for parameter $\theta_i$ is defined as

$$
\phi_i = \sum_{J \subseteq \{1, \ldots, m\} \setminus \{i\}} \frac{(m - |J| - 1)! \, |J|!}{m!} \left[ c(J \cup \{i\}) - c(J) \right],
\tag{8}
$$

for $i \in \{1, 2, \ldots, m\}$, where $j$ ranges over all subsets of parameter indices excluding $i$, and $c(J)$ denotes a cost function. For sensitivity analysis, we define the cost function as $c(J) := \mathbb{E}[\text{Var}(y \mid \theta_{\sim J})]$, which quantifies the expected remaining variance of the model output when the parameters not in $j$ are known. The term $c(J \cup \{i\}) - c(J)$ captures the incremental reduction in unexplained variance attributable to parameter $\theta_i$, given that the parameters in $j$ are already known. This specific choice

of the cost function is motivated by the fact that its Monte Carlo estimator remains unbiased regardless of the sample size, making it numerically efficient for estimating $\phi_i$ [7]. The Shapley effect allocates the output variance among input parameters by considering all possible subsets of model inputs. It guarantees that the sensitivity indices for each parameter sum to one.

Both Rosenblatt transformations and Shapley effects offer theoretically rigorous methods for handling correlations within $\boldsymbol{\theta}$. However, the numerical estimation of these indices remains challenging. From the definitions of the sensitivity indices in (7) and (8), it is clear that both approaches involve conditional expectations of the model output $y$ given certain parameter values. Consequently, Monte Carlo estimators for these indices require accurate sampling from conditional distributions. The Rosenblatt transformation method requires the numerical approximation of $m$ transformations, each corresponding to a specific permutation order, to generate conditional samples for sensitivity analysis. This is typically implemented by transforming independent uniform samples into parameter space using the inverse conditional CDFs. On the other hand, evaluating Shapley effects requires conditional sampling over all $2^m$ subsets of input parameters, which is computationally intensive and highly non-trivial, especially for high-dimensional models.

Existing conditional sampling methods, such as copula based approaches, are widely used for estimating global sensitivity indices when parameters are correlated. Techniques like ARTA [19] and NORTA [20] transform correlated Gaussian variables into arbitrary marginal distributions while preserving a specified correlation structure. Although computationally efficient and flexible in low-dimensional settings, these methods assume linear correlations and symmetry in the dependence structure, which limits their ability to capture non-Gaussian or non-linear dependencies often present in complex models [21]. To avoid biased conditional samples and inaccurate sensitivity estimates, more flexible methods are required to effectively capture complex dependencies among $\boldsymbol{\theta}$.

## Methods

The process begins with calibrating the mathematical model (1) through Bayesian inference using observed data. As discussed in Bayesian Inference section, we obtain a posterior sample set $S$ via standard MCMC algorithms, which includes all the information available given the data $D$ and the model structure. Based on the discussion in Handling Correlated Parameters section, both the Rosenblatt transformation and Shapley effects provide theoretically sound approaches for conducting GSA using posterior samples. The key challenge lies in accurately modeling the potentially high-dimensional and complex dependencies within the posterior distribution and generating high quality conditional samples to facilitate Monte Carlo estimation of the sensitivity indices.

It is important to note that Posterior samples generated by MCMC are inherently dependent [22], which can affect the training efficiency of generative models. Strong autocorrelation reduces the effective sample size and introduces redundancy in the training data, potentially slowing convergence. This dependence impacts statistical efficiency rather than the validity of the learning objective. When generative models are trained via maximum likelihood (or its evidence lower bound variant), consistency is preserved provided the MCMC chain is ergodic and has reached stationarity [23,24]. To mitigate dependence effects, we apply standard preprocessing steps, including discarding burn-in samples and thinning the chain by retaining every 20th draw to reduce short range autocorrelation. In practice, generative models trained on moderately thinned posterior samples converge reliably and accurately recover both joint and conditional posterior distributions, as demonstrated in our numerical experiments and in the relevant study [25]. These results indicate that, with appropriate diagnostics and preprocessing, MCMC dependence does not compromise the reliability of the learned distributions and the resulting sensitivity index estimates.

### Autoregressive Model

Autoregressive models constitute a powerful class of generative models for high-dimensional probability distributions. This line of research originated with the proposal of Non-linear Independent Components Estimation (NICE), and was subsequently enhanced by developments in Neural Autoregressive Distribution Estimation (NADE) and Masked Autoencoder

for Distribution Estimation (MADE) [26]. These methods represent the posterior distribution of the parameter vector $\theta = (\theta_1, \theta_2, \ldots, \theta_m)$ as a product of one-dimensional conditional distributions

$$p(\theta \mid D) = \prod_{j=1}^{m} p(\theta_j \mid \theta_{<j}, D),$$

where $\theta_{<j}$ denotes the preceding parameters $(\theta_1, \ldots, \theta_{j-1})$. Each conditional distribution is modeled as a Gaussian distribution, with neural networks predicting the mean and log-standard deviation

$$p(\theta_j \mid \theta_{<j}, D) = \mathcal{N}(\theta_j \mid m_j(\theta_{<j}, D), \sigma_j^2(\theta_{<j}, D)). \tag{9}$$

To preserve the autoregressive structure, the MADE framework is commonly employed, which utilizes masked neural networks to ensure that each output depends only on preceding input variables. This approach has been demonstrated to be equivalent to implementing a single layer normalizing flow that transforms a simple standard normal distribution into the target complex density, as formalized in Masked Autoregressive Flow (MAF) [27].

To design a MAF where the base distribution is an independent uniform distribution $\mathcal{U}[0, 1]^m$ corresponding to a Rosenblatt transformation (6), we must develop an invertible transformation between an $m$-dimensional uniform random vector $z$ and the target model parameter $\theta$. This transformation is based on the masked neural network structure in equation (9) that generates the autoregressive structure corresponding to the exact permutation order in the Rosenblatt transformation. The implementation involves designing both a forward transformation that converts uniform samples to posterior samples for the training process, and the corresponding inverse transformation for generating samples from the posterior distribution.

The forward transformation $z = f_\phi(\theta)$ maps each posterior sample $\theta \in \mathbb{R}^m$ to a latent variable $z \sim \mathcal{U}[0, 1]^m$ using the standard Gaussian CDF

$$z_j = \Phi\left(\frac{\theta_j - m_j(\theta_{<j}, D)}{\sigma_j(\theta_{<j}, D)}\right), \quad j = 1, \ldots, m,$$

where $\Phi(\cdot)$ is the standard normal CDF, and $m_j, \sigma_j$ are outputs of an neural network conditioned on the preceding components $\theta_{<j}$. To train the model, we maximize the likelihood of observing the posterior samples. Since the latent distribution is uniform and has zero log-density, the log-likelihood is determined solely by the Jacobian of the transformation. Due to the autoregressive structure, the Jacobian is lower triangular, and its log-determinant simplifies to a sum over the diagonal elements

$$\log\left|\det\left(\frac{\partial f_\phi}{\partial \theta}\right)\right| = \sum_{j=1}^{m}\left[\log\varphi\left(\frac{\theta_j - m_{j,\phi}(\theta_{<j}, D)}{\sigma_{j,\phi}(\theta_{<j}, D)}\right) - \log\sigma_{j,\phi}(\theta_{<j}, D)\right],$$

where $\varphi(\cdot)$ denotes the standard normal probability density function (PDF). In practice, the total log-likelihood over a posterior sample set $S$ is maximized during training using this expression. After training process, new samples can be generated by sampling from latent distribution $\mathcal{U}[0, 1]^m$ and applying the inverse transformation autoregressively

$$\theta_j = m_{j,\phi}(\theta_{<j}, D) + \sigma_{j,\phi}(\theta_{<j}, D) \cdot \Phi^{-1}(z_j), \quad j = 1, \ldots, m,$$

where $\Phi^{-1}(\cdot)$ is the inverse standard normal CDF.

Although autoregressive models provide a simple and effective framework for learning transformations between random variables, their performance is highly sensitive to the ordering of input variables, particularly when some marginal

distributions are non-Gaussian [27]. In the case of the Rosenblatt transformation (6), the variable ordering must be fixed to a specific natural cyclic permutation, which limits its applicability to more complex models. Additionally, modeling each conditional distribution with a single Gaussian may constrain the model's ability to capture complex dependencies. A more expressive alternative is to use a mixture of Gaussians for each conditional distribution

$$p(\theta_j \mid \boldsymbol{\theta}_{<j}, D) = \sum_{k=1}^{K} \pi_{j,k}(\boldsymbol{\theta}_{<j}, D) \, \mathcal{N}\left(\theta_j \mid \mu_{j,k}(\boldsymbol{\theta}_{<j}, D), \sigma_{j,k}^2(\boldsymbol{\theta}_{<j}, D)\right),$$

where the mixture weights $\pi_{j,k}$, means $\mu_{j,k}$, and standard deviations $\sigma_{j,k}$ are all predicted by a neural network. However, a key limitation of this approach is that Gaussian mixtures are not analytically invertible. Consequently, the model can no longer be considered a well defined normalizing flow, and generating $\boldsymbol{\theta}$ from a latent variable $\boldsymbol{z}$ becomes a stochastic process. This introduces randomness into the sampling procedure, which is undesirable for estimating sensitivity indices using the two-matrix estimator (5), where deterministic transformations are preferred.

Despite these limitations, the Autoregressive model still provides a simple and efficient framework for estimating Sobol' sensitivity indices (7), as we will demonstrate in the numerical experiments. When estimating all four Sobol' indices for a correlated parameter vector $\boldsymbol{\theta}$, it is necessary to approximate $m$ Rosenblatt transformations, each corresponding to a different permutation of the input parameters. To accomplish this, we train $m$ separate Autoregressive models, one for each permutation. The data pipeline for training each autoregressive model begins with $N$ posterior samples obtained from Bayesian calibration. For each Rosenblatt permutation, a separate training dataset is constructed by reordering the parameter components accordingly. Each conditional density is parameterized using a MADE-style masked multilayer perceptron (MLP) with a specified number of hidden layers and neurons per layer, employing ReLU activation functions to enforce the autoregressive dependency structure. The output layer predicts the conditional mean and log-standard deviation for each parameter component. Masking is implemented following the standard MADE construction to ensure the prescribed variable ordering. Model training is performed by maximizing the log-likelihood of the posterior samples. All models are implemented in PyTorch, building upon the open source MADE architecture provided in [28] for masked autoregressive flows, with custom modifications to accommodate a uniform base distribution corresponding to the Rosenblatt transformation. The full implementation is publicly available at [29]. The following algorithm 1 summarizes the full procedure.

### Algorithm 1 GSA with Autoregressive model

1. **Input:** Posterior samples $S \in \mathbb{R}^{N \times m}$, model output $g : \mathbb{R}^m \to \mathbb{R}$, Monte Carlo sample size $N_S$
2. **Output:** Sobol' indices $\{\widehat{S}_j^{\text{full}}, \widehat{S}_{T_j}^{\text{full}}, \widehat{S}_j^{\text{ind}}, \widehat{S}_{T_j}^{\text{ind}}\}_{j=1}^{m}$
3. Compute the sample mean and variance based on posterior predictive outputs

$$\bar{y} = \frac{1}{N} \sum_{i=1}^{N} g(\theta_i), \quad \widehat{V} = \frac{1}{N-1} \sum_{i=1}^{N} \left(g(\theta_i) - \bar{y}\right)^2$$

4. **for** $j = 1$ to $m$ **do**
5.   Define cyclic permutation $\pi_j$ with index $j$ as the first coordinate
6.   Train a MADE model with autoregressive ordering $\pi_j$ on $S$
7.   Generate $S_A, S_B \sim \mathcal{U}[0, 1]^m$ of size $N_S$
8.   Create hybrid $S_{AB_j}$ by replacing the first column of $S_A$ with that of $S_B$
9.   Create hybrid $S_{AB_{j-1}}$ by replacing the last column of $S_A$ with that of $S_B$
10.  Apply the inverse MADE transformation to obtain:
11.    $\boldsymbol{\theta}^{(A)}$, $\boldsymbol{\theta}^{(B)}$, $\boldsymbol{\theta}^{(AB_j)}$, and $\boldsymbol{\theta}^{(AB_{j-1})}$
12.  Evaluate the model output
13.    $\boldsymbol{y}^{(A)} = g(\boldsymbol{\theta}^{(A)})$, $\boldsymbol{y}^{(B)} = g(\boldsymbol{\theta}^{(B)})$, $\boldsymbol{y}^{(AB_j)} = g(\boldsymbol{\theta}^{(AB_j)})$, $\boldsymbol{y}^{(AB_{j-1})} = g(\boldsymbol{\theta}^{(AB_{j-1})})$
14.    Estimate full first order and total order Sobol' indices

15. $$\widehat{S}_j^{\texttt{full}} = \frac{\frac{1}{N_S}\sum_{i=1}^{N_S} y_i^{(AB_j)}(y_i^{(AB_j)}-y_i^{(A)})}{\widehat{v}}, \quad \widehat{S}_{T_j}^{\texttt{full}} = \frac{\frac{1}{N_S}\sum_{i=1}^{N_S}(y_i^{(AB_j)}-y_i^{(A)})^2}{2\widehat{v}}$$

Estimate independent first order and total order Sobol' indices

$$\widehat{S}_{j-1}^{\texttt{ind}} = \frac{\frac{1}{N_S}\sum_{i=1}^{N_S} y_i^{(AB_j-1)}(y_i^{(AB_j-1)}-y_i^{(A)})}{\widehat{v}}, \quad \widehat{S}_{T_{j-1}}^{\texttt{ind}} = \frac{\frac{1}{N_S}\sum_{i=1}^{N_S}(y_i^{(AB_j-1)}-y_i^{(A)})^2}{2\widehat{v}}$$

16. **end for**

17. **Return** $\{\widehat{S}_j^{\texttt{full}}, \widehat{S}_{T_j}^{\texttt{full}}, \widehat{S}_j^{\texttt{ind}}, \widehat{S}_{T_j}^{\texttt{ind}}\}_{j=1}^m$

## Diffusion Model

Diffusion models provide a powerful alternative for modeling the posterior distribution from a given sample set. In the context of GSA with Shapley effects, diffusion models are more flexible and capable than Autoregressive models due to their ability to generate arbitrarily conditioned samples. We begin by considering classical Denoising Diffusion Probabilistic Models (DDPMs) [21]. These models learn to approximate the posterior by reversing a diffusion process that gradually transform the calibrated model parameters into pure Gaussian noise over $T$ steps. The forward process at each time step $t$ is defined as

$$q(\theta_t \mid \theta_{t-1}) = \mathcal{N}(\theta_t; \sqrt{1-\beta_t}\,\theta_{t-1}, \beta_t \boldsymbol{I}), \quad t = 1, \ldots, T,$$

where $\beta_t$ is a small, fixed variance that controls the noise added at each step. Using the Markov property, we can directly write the distribution of $\theta_t$ given the initial posterior sample $\theta_0 \sim p(\theta \mid D)$ as

$$q(\theta_t \mid \theta_0) = \mathcal{N}(\theta_t; \sqrt{\bar{\alpha}_t}\,\theta_0, (1-\bar{\alpha}_t)\boldsymbol{I}),$$

where $\alpha_t = 1 - \beta_t$ and $\bar{\alpha}_t = \prod_{s=1}^t \alpha_s$. The DDPM is trained to reverse the diffusion process. Given a noisy sample $\theta_t$, a neural network is used to predict the denoised $\theta_{t-1}$ through

$$p_\phi(\theta_{t-1} \mid \theta_t) = \mathcal{N}(\theta_{t-1}; \mu_\phi(\theta_t, t), \Sigma_\phi(\theta_t, t)),$$

where $\mu_\phi$ and $\Sigma_\phi$ denote the mean and covariance functions parameterized by a neural network. A common choice for parameterizing the mean and covariance is

$$\mu_\phi(\theta_t, t) = \frac{1}{\sqrt{\alpha_t}}\left(\theta_t - \frac{\beta_t}{\sqrt{1-\bar{\alpha}_t}}\epsilon_\phi(\theta_t, t)\right), \quad \Sigma_\phi(\theta_t, t) = \sigma_t^2 \boldsymbol{I},$$

where $\epsilon_\phi(\theta_t, t)$ predicts the noise that was added to $\theta_0$, and $\sigma_t^2$ is typically a fixed or scheduled variance. The training objective is to maximize the likelihood of the original posterior sample $\theta_0 \sim p(\theta \mid D)$. Since the true reverse transitions $q(\theta_{t-1} \mid \theta_t)$ are intractable, training instead maximizes a variational lower bound on the log-likelihood, known as the evidence lower bound (ELBO), to approximate the true posterior. In practice, this is often simplified to a denoising score matching loss, which is equivalent to a weighted form of the ELBO [30]. The idea is to recover the noise $\epsilon \sim \mathcal{N}(\boldsymbol{0}, \boldsymbol{I})$ from a noisy sample generated via

$$\theta_t = \sqrt{\bar{\alpha}_t}\,\theta_0 + \sqrt{1-\bar{\alpha}_t}\,\epsilon.$$

Then the noise prediction network $\epsilon_\phi(\theta_t, t)$ is trained by minimizing the expected squared error

$$\mathcal{L} = \mathbb{E}_{\theta_0, \epsilon, t}\left[\|\epsilon - \epsilon_\phi(\theta_t, t)\|^2\right].$$

DDPMs have proven effective at modeling high-dimensional and complex distributions, as evidenced by recent advances in large scale machine learning. For very high-dimensional settings, U-Net architectures are commonly employed to enhance performance [30]. In contrast, for moderately sized parameter spaces, a standard MLP often suffices.

With a trained DDPM that accurately captures the posterior $p(\theta \mid D)$, we turn to the task of generating conditional samples given an arbitrary subset of parameters $\theta_J \subsetneq \theta$. This task is commonly referred to as the inpainting problem in computer vision. There are two main approaches for conditional generation. The first trains the diffusion model jointly with the conditional information and performs denoising with conditioning. A well known example is classifier-free guidance [31]. The second approach uses a DDPM trained on the joint distribution and designs a procedure to generate conditional samples without additional training. An example of this is the RePaint algorithm [32].

In our case, sensitivity index estimation requires conditional samples for arbitrary parameter subvector $\theta_J$. Training a diffusion model for each possible conditioning is computationally infeasible. We therefore adopt the RePaint method to generate conditional samples without requiring retraining. To sample from the conditional distribution $p(\theta \mid \theta_J)$, the denoising process proceeds as follows. At each reverse step, we first sample the unconditional part from the learned reverse process

$$\theta_{t-1}^{(\sim J)} \sim p_\phi(\theta_{t-1} \mid \theta_t).$$

The conditional part is then resampled by noising the fixed parameters $\theta_J$ to the same level of diffusion at step $t-1$

$$\theta_{t-1}^{(J)} \sim \mathcal{N}(\sqrt{\bar{\alpha}_{t-1}}\theta_J, (1-\bar{\alpha}_{t-1})I).$$

Both parts are combined using element-wise multiplication with a binary mask $m$ that indicates the positions of the fixed parameters

$$\theta_{t-1} = m \odot \theta_{t-1}^{(J)} + (1-m) \odot \theta_{t-1}^{(\sim J)}.$$

To further improve consistency between the conditioned and unconditioned components, RePaint reintroduce noise after merging by sampling from the forward process

$$\theta_t' \sim \mathcal{N}(\sqrt{1-\beta_t}\theta_{t-1}, \beta_t I).$$

Allowing multiple resampling rounds helps gradually align the samples with the desired conditional distribution. The DDPM, combined with the RePaint method, provides a powerful approach for generating conditional samples given arbitrary values on any parameter subset. This makes it particularly suitable for estimating Shapley effects sensitivity indices (8). Since the Shapley effect quantifies the average contribution of a parameter across all possible subsets, conditional samples must be generated for $2^m$ subsets, where $m$ is the number of parameters. With each subset, a number of outer samples $N_O$ and inner samples $N_I$ are required to estimate the cost function $c(J) = \mathbb{E}(\text{Var}[y \mid \theta_{\sim J}])$, which quickly becomes computationally infeasible.

Fortunately, a computationally efficient alternative known as the random permutation method has been proposed to approximate the Shapley effects [7]. This method estimates the effect as

$$\phi_i = \frac{1}{M} \sum_{k=1}^{M} \left[ c(P_i(\pi_k) \cup \{i\}) - c(P_i(\pi_k)) \right],$$

(10)

for $i = 1, 2, \cdots, m$, where $P_i(\pi_k)$ denotes the set of parameters that precede parameter $i$ in a random permutation $\pi_k \in \Pi(m)$. For simplicity of notation, we use $P_{i+1}(\pi_k) := P_i(\pi_k) \cup \{i\}$, and $-P_{i+1}(\pi_k)$ as the set of parameters that follow parameter $i$ (excluding $i$) in the permutation $\pi_k$. It should be noted that during the estimation of the Shapley effect, the independent total order index $S_{T_i}^{\text{ind}}$ and the full first order Sobol index $S_i^{\text{full}}$ are estimated simultaneously when $P_i(\pi_k) = \emptyset$ and $-P_{i+1}(\pi_k) = \emptyset$, respectively. E. Song et al. [7] have shown that the optimal configuration for this algorithm is $N_O = 1$, $N_I = 3$, and $M$ as large as possible. Algorithm 2, adapted from Algorithm 1 in E. Song [7], outlines the detailed procedure for applying diffusion models to estimate sensitivity indices.

### Algorithm 2 GSA with a Diffusion Model

1. **Input:** Posterior samples $S \in \mathbb{R}^{N \times m}$, model output $g : \mathbb{R}^m \longrightarrow \mathbb{R}$, Initial sample size $N_V$, number of permutations $M$, outer sample size $N_O$, inner sample size $N_I$
2. **Output:** Sensitivity indices $\{\widehat{\phi}_j\}_{j=1}^m$
3. Initialize $\widehat{\phi}_j = 0$ for $j = 1, 2, \ldots, m$ and $k = 0$
4. Compute posterior sample mean and variance of model outputs from $S$

$$\bar{y} = \frac{1}{N_V} \sum_{i=1}^{N_V} g(\theta_i), \quad \widehat{V} = \frac{1}{N_V - 1} \sum_{i=1}^{N_V} \left( g(\theta_i) - \bar{y} \right)^2$$

5. Train a diffusion model on $S$ to learn the augmented posterior $p(\boldsymbol{\theta} \mid D)$
6. **while** $k < M$ **do**
7. Draw a random permutation $\pi_k \in \Pi(m)$
8. Set $C = 0$
9. **for** $j = 1, 2, \ldots, m$ **do**
10. **if** $j = m$ **then**
11. $\widehat{c}(P_{\pi(j+1)}(\pi)) = \widehat{V}$ $\triangleright$ Comment: $P_{\pi(j+1)}(\pi) = P_{\pi(j)}(\pi) \cup \pi(j)$
12. **else**
13. Draw $N_O$ samples $\{\boldsymbol{\theta}_{-P_{\pi(j+1)}(\pi)}^{(1)}\}_{l=1}^{N_O}$ from $S$
14. **for** $l = 1$ to $N_O$ **do**
15. Generate $N_I$ samples $\{\boldsymbol{\theta}^{(1,h)}\}_{h=1}^{N_I}$ from $p(\boldsymbol{\theta} \mid \boldsymbol{\theta}_{-P_{\pi(j+1)}(\pi)}^{(1)})$ via RePaint
16. Evaluate $y^{(1,h)} = g\left(\boldsymbol{\theta}^{(1,h)}\right)$
17. **end for**
18. Estimate means and conditional variances

$$\bar{y}^{(1)} = \frac{1}{N_I} \sum_{h=1}^{N_I} y^{(1,h)}, \quad \widehat{v}^{(1)} = \frac{1}{N_I - 1} \sum_{h=1}^{N_I} \left( y^{(1,h)} - \bar{y}^{(1)} \right)^2$$

19. Estimate cost $\widehat{c}(P_{\pi(j+1)}(\pi)) = \frac{1}{N_O} \sum_{l=1}^{N_O} \widehat{v}^{(1)}$
20. **end if**
21. Compute $\widehat{\Delta}_{\pi(j)}(\pi) = \widehat{c}(P_{\pi(j+1)}(\pi)) - C$
22. Update $\widehat{\phi}_{\pi(j)} = \widehat{\phi}_{\pi(j)} + \widehat{\Delta}_{\pi(j)}(\pi)$
23. Set $C = \widehat{c}(P_{\pi(j+1)}(\pi))$
24. **end for**
25. $k = k + 1$
26. **end while**
27. **Return:** Final sensitivity indices $\widehat{\phi}_j = \frac{\widehat{\phi}_j}{M}$ for $j = 1, 2, \ldots, m$

## Case Study Results

We present three case studies to demonstrate the effectiveness and applicability of our method across mathematical models of varying complexity, with the latter two inspired by health science research problems. In the Benchmark Study section, we evaluate the efficiency and accuracy of the proposed methods relative to traditional GSA methods across varying input correlations and dimensions. In the Epidemiology Model section, we apply the autoregressive model to a practical problem in infectious disease modeling based on the classical Susceptible-Infectious-Recovered (SIR) framework. Finally, in the Cellular Dynamic Model section, we consider a more complex model describing the dynamics of

chimeric antigen receptor (CAR)-T cell therapy for blood cancer treatment, proving the effectiveness and efficiency of our GSA approach using diffusion models.

## Benchmark Study

In this section, we present a benchmark study to compare the performance of different GSA methods for estimating the full total order sensitivity index $S_{T_i}^{\text{full}}$. Among traditional approaches, we consider the classical Sobol two-matrix estimator (5), which is designed for models with independent inputs, as well as two widely used improved methods. The first is a copula based method [20], which models input dependencies by combining marginal distribution estimation with a Gaussian copula to capture linear correlations among inputs. The second is the Polynomial Chaos Expansion (PCE) method [33], which approximates the input–output relationship using a PCE surrogate and computes sensitivity indices analytically from the estimated expansion coefficients to improve the computational efficiency. We compare these traditional methods with the proposed generative model based approaches in terms of both accuracy and computational efficiency. To enable reliable quantitative comparisons, we employ standard benchmark functions with known analytical solutions for the total order sensitivity indices, including a linear function, the Ishigami function, and a high-dimensional Sobol $g$-function. These benchmarks allow for systematic and reproducible evaluation. Furthermore, we consider a range of input distributions to investigate the effects of increasing input correlation complexity and dimensionality on the performance of both traditional GSA methods and the proposed generative model based approaches. The implementations of the copula and PCE methods are carried out using the UQLab software package [34].

  **Linear Function.** Variance based sensitivity indices for simple linear functions such as $y = X_1 + X_2 + X_3$ have been extensively studied and are commonly used as benchmarks for evaluating the accuracy of sensitivity index estimators. In particular, the analytical total order sensitivity index $S_{T_i}^{\text{full}}$ for the case where $\boldsymbol{X} = (X_1, X_2, X_3)$ follows a correlated Gaussian distribution, $\boldsymbol{X} \sim \mathcal{N}(\boldsymbol{\mu}, \Sigma)$, have been derived in [35]. In this example, we compare the theoretical values of the sensitivity indices with those estimated using traditional and proposed GSA methods. To set up the linear function, we assume the true distribution of the input vector follows a multivariate Gaussian distribution $\mathcal{N}(\boldsymbol{\mu}, \Sigma)$, where

$$\boldsymbol{\mu} = \boldsymbol{0}, \quad \underline{\Sigma} = \begin{bmatrix} 0.04 & 0.06 & 0.10 \\ 0.06 & 0.36 & 0.30 \\ 0.10 & 0.30 & 1.00 \end{bmatrix}.$$

The covariance matrix $\Sigma$ can be arbitrarily chosen, and we select this particular form to simplify the analytical computation of the sensitivity indices.

  The GSA methods considered in this benchmark study include the classical Sobol two-matrix estimator, the Gaussian copula method, the PCE method, and the two proposed generative model based approaches described in Algorithms 1 and 2. All methods rely on sampling from the input distribution to estimate sensitivity indices. In general, sampling serves two purposes. First, samples drawn from the joint input distribution are used to train a surrogate model that captures the underlying dependence structure among the inputs. Second, samples generated from the trained surrogate are used to produce conditional samples for estimating sensitivity indices. We denote by $N_1$ the sample size used for surrogate model training, and by $N_2$ the number of samples used for sensitivity index estimation. Two methods constitute notable exceptions. For the classical Sobol two-matrix estimator, no surrogate model is constructed, and therefore $N_1 = 0$. In contrast, for the PCE method, sensitivity indices are evaluated analytically from the estimated PCE coefficients, and thus no additional sampling is required for sensitivity estimation, i.e., $N_2 = 0$.

  To set up the benchmark study, we first specify the surrogate model structure for each method. For the Gaussian copula method, commonly used distribution families such as Gaussian, Uniform, or Beta distributions are employed to approximate the marginal distributions, while the dependence structure is captured by a correlation matrix. For the PCE method,

a polynomial chaos surrogate of maximum degree $p = 2$ is trained to approximate the true model. For the proposed generative model based approaches, the surrogate input distributions are learned using neural networks with standard MLP architectures consisting of two hidden layers with 32 neurons per layer. These architectures are used for both the masked network in the autoregressive model and the noise prediction network in the diffusion model. For a linear function with three-dimensional inputs, we empirically choose the sample sizes $N_1 = 6,000$ and $N_2 = 2,000$, where $N_2$ corresponds to $N_S$ in Algorithm 1 and to $M \times N_I \times M_O$ in Algorithm 2. This choice is empirical but sufficient for the present low-dimensional setting. The impact of the training sample size and the effective Monte Carlo sample size is further investigated in the following sections. To improve estimation stability and quantify uncertainty in the sensitivity indices, a bootstrap procedure is employed. For methods that rely on Monte Carlo sampling, 2,000 bootstrap replicates of size $N_2$ are used. For the PCE method, where $N_2 = 0$, confidence intervals are obtained by bootstrapping the training dataset used to estimate the PCE coefficients and recomputing the sensitivity indices for each replicate. In all cases, the mean of the bootstrap estimates is reported as the final result, and detailed bootstrap uncertainty plots are provided in Fig B in S1 Appendix.

We now perform GSA using all methods discussed above for the linear function, and compare the resulting sensitivity index estimates with their corresponding analytical values. Table 1 summarizes the results of this comparison.

Based on Table 1, we observe that the traditional two-matrix estimator and PCE method exhibit significant bias compared to the theoretical total order index values. This is primarily because both methods rely on the assumption of independent input distributions. The two-matrix estimator achieves control over the conditioning variables by switching columns in two sample matrices, while PCE assumes input independence for index computation. The presence of correlation violates this assumption, leading to inaccurate estimates. In Fig A in S1 Appendix, we provide a comparison of scatter plots for the true input distribution and those generated through column swapping. These plots clearly demonstrate that the correlation structure is distorted by the swapping procedure. In contrast, the Gaussian copula method and the generative model based methods provide accurate estimates. This suggests that both the copula and generative models accurately capture the original correlated input distribution, thereby yielding more reliable sensitivity index estimates. This finding emphasizes that input correlation significantly impacts estimation results and must be considered when present. We will next explore a case where the underlying correlations are complex, demonstrating that a Gaussian copula may be insufficient, and a generative model is necessary to capture the actual input distribution for correct estimation.

**Ishigami Function.** In addition to the simple linear function, the Ishigami function [36] is another widely used benchmark for testing the robustness of GSA methods, particularly in the presence of strong nonlinearity and interaction effects among inputs. The three-dimensional Ishigami function is defined as

$$Y = \sin(X_1) + a\sin^2(X_2) + bX_3^4\sin(X_1),$$

(11)

where $\boldsymbol{X} = (X_1, X_2, X_3)$ denotes the input vector and $a$ and $b$ control the magnitude of nonlinear and interaction effects. Throughout this study, we set $a = 7$ and $b = 0.1$.

To emulate a Bayesian-calibrated input distribution, we consider a non-Gaussian input distribution with asymmetric and tail dependent correlations. Such dependency structures are commonly observed in practice and are not adequately represented by Gaussian copulas. The marginal distributions of the inputs are specified as

**Table 1. Comparison of full total order sensitivity indices for the linear function across different GSA methods. Estimates are reported as bootstrap means.**

| Index | Sobol | Copula | PCE | AR | Diffusion | Analytical |
|---|---|---|---|---|---|---|
| $S_{T_1}^{full}$ | 0.03 | 0.01 | 0.05 | 0.01 | 0.01 | 0.01 |
| $S_{T_2}^{full}$ | 0.25 | 0.10 | 0.29 | 0.09 | 0.09 | 0.10 |
| $S_{T_3}^{full}$ | 0.73 | 0.28 | 0.68 | 0.30 | 0.29 | 0.29 |

$$X_1 \sim \mathcal{N}(0, 1), \quad X_2 \sim \text{Beta}(10, 2), \quad X_3 \sim \text{Beta}(2, 10),$$

where $X_2$ is left skewed and $X_3$ is right skewed. The dependence structure among the inputs is modeled using a canonical vine (C-vine) copula, which constructs a multivariate distribution through a sequence of bivariate copulas. We select $X_1$ as the canonical node, and define a two level dependence structure.

At the first level, unconditional dependencies are defined between $X_1$ and the remaining variables. The pair $(X_1, X_2)$ is modeled using a Student-$t$ copula with one degree of freedom, capturing strong symmetric tail dependence. The dependence between $(X_1, X_3)$ is modeled using a Gumbel copula, which exhibits upper tail dependence and introduces asymmetry into the joint distribution. At the second level, the conditional dependence between $X_2$ and $X_3$ given $X_1$ is modeled using a Clayton copula rotated by 180°. This choice induces strong lower tail dependence in the conditional distribution, further increasing the complexity of the input dependence structure. Overall, the combination of non-Gaussian marginals and heterogeneous pair copula families provides a challenging test case for GSA methods under nonlinear and asymmetric dependencies.

For this input distribution, the theoretical total order sensitivity indices are estimated though Monte Carlo simulation using the known joint distribution. All GSA methods considered in Linear Function section are then applied using the same sample size, hyperparameters, and neural network architecture, as the input dimension remains unchanged. The estimated sensitivity indices are compared with the theoretical values, and the results are summarized in Table 2. From Table 2, we observe that methods relying on independence or Gaussian copula assumptions, including the classical two-matrix estimator, the PCE approach, and Gaussian copula based methods, exhibit substantial bias. The two-matrix and PCE methods assume independent inputs and therefore show the largest discrepancies when applied to this strongly dependent and non-Gaussian input model. The autoregressive surrogate model partially mitigates sampling bias by learning input dependencies directly from data. However, its representational capacity is limited when the input distribution exhibits skewed marginals and heavy-tailed copulas because a single Gaussian cannot capture asymmetric conditional dependencies, leading to persistent bias in the estimated sensitivity indices. In contrast, the diffusion based model consistently yields the most accurate sensitivity index estimates, closely matching the theoretical values. This superior performance underscores the importance of highly flexible generative models for GSA under complex, non-Gaussian dependence structures. A comprehensive comparison of the true input distribution, along with the conditional distributions used for estimating sensitivity indices, and their approximations obtained via the Gaussian copula, the autoregressive model, and the diffusion based model is provided in S2 Appendix.

We have implemented the same neural network structure and retained the training sample size $N_1 = 6,000$ as in the linear function study. However, we did not discuss the optimal combination of network architecture and the corresponding minimal training sample size required for the generative model to perform effectively. In S2 Appendix, we present a performance comparison of the generative models in recovering the true input distributions across different groupings, using varying numbers of training samples and neural network architectures. The results indicate that current settings fall within the optimal range for accurately modeling such a complex input distribution.

**Table 2. Comparison of full total order sensitivity indices for the Ishigami function across different GSA methods. Estimates are reported as bootstrap means.**

| Index | Sobol | Copula | PCE | AR | Diffusion | Analytical |
|---|---|---|---|---|---|---|
| $S_{T_1}^{\text{full}}$ | 0.84 | 0.10 | 0.74 | 0.08 | 0.07 | 0.06 |
| $S_{T_2}^{\text{full}}$ | 0.01 | 0.32 | 0.37 | 0.19 | 0.23 | 0.24 |
| $S_{T_3}^{\text{full}}$ | 0.62 | 0.00 | 0.62 | 0.09 | 0.02 | 0.02 |

**Sobol Function.** We have compared the performance of traditional and proposed GSA methods under low-dimensional input settings with distributions of varying complexity. In practical applications, however, high input dimensionality poses an additional challenge, as it exacerbates the curse of dimensionality in surrogate modeling and imposes a substantial computational burden when estimating sensitivity indices through Monte Carlo stimulation. To address this issue, we design a numerical benchmark to evaluate and compare the computational performance of different GSA approaches in a high-dimensional setting. The Sobol $g$-function [37] is a standard benchmark for assessing sensitivity analysis methods in high dimensions. For a model with $m$ inputs, it is defined as

$$Y = \prod_{i=1}^{m} \frac{|4X_i - 2| + a_i}{1 + a_i}, \quad i = 1, 2, \ldots, m,$$

(12)

where $\boldsymbol{a} = (a_1, a_2, \ldots, a_m)$ are non-negative parameters controlling the relative importance of each input variable. Assuming the input vector $\boldsymbol{X} = (X_1, X_2, \ldots, X_m)$ is uniformly distributed over the unit hypercube $[0, 1]^m$, the variance decomposition admits closed-form expressions. In particular, the total order sensitivity index of input $X_i$ is given by

$$S_{T_i}^{\text{full}} = \frac{V_i \Pi_{j \neq i}(1 + V_j)}{\prod_{j=1}^{m}(V_j + 1) - 1}, \qquad V_i = \frac{1}{3(1 + a_i)^2},$$

(13)

for $i = 1, 2, \ldots, m$. This analytical tractability makes the Sobol $g$-function well suited for benchmarking GSA methods.

We set the input dimension to $m = 200$. The parameters are specified as $a_i = (i - 1)/10$ for $1 \leq i \leq 20$ and $a_i = 99$ for $21 \leq i \leq m$. Under this configuration, only the first 20 input variables exert a non-negligible influence on the model output, while the remaining inputs contribute negligibly. We apply all GSA methods introduced in the previous sections to estimate the total order sensitivity indices and compare the results against the corresponding analytical reference values. Since practical interest typically lies in the influential inputs, we restrict the index comparison to those inputs with theoretical total order sensitivity indices exceeding 0.1. The overall estimation accuracy is additionally quantified using the mean of absolute error (MAE),

$$\text{MAE} = \frac{1}{m} \sum_{i=1}^{m} \left| S_i^{\text{full}} - \widehat{S}_i^{\text{full}} \right|.$$

As the number of input variables increases substantially compared to the previous example, both the surrogate training sample size $N_1$ and the Monte Carlo sample size $N_2$ must be increased accordingly. For the PCE surrogate with a maximum polynomial degree $p$, the number of coefficients to be estimated is $\binom{m+p}{p}$, while the Gaussian copula approach requires $m(m + 1)/2$ parameters to construct the correlation matrix. The number of parameters required by these methods therefore increases rapidly with the input dimension $m$, quickly exceeding that of the default two-layer neural network with 32 neurons used in both the autoregressive model and the noise prediction network in the diffusion model. We show that the relatively simple neural network architecture employed in the previous three-dimensional problems can still achieve strong performance in this high-dimensional setting. To ensure accurate training of all surrogate models while mitigating the risk of overfitting, we set the surrogate training sample size to $N_1 = 1 \times 10^5$. The Monte Carlo sample size for estimating the sensitivity indices is set to $N_2 = 20,000$ for this high-dimensional problem. The resulting GSA estimates are reported in Table 3.

As shown in Table 3, all considered methods are able to recover the sensitivity structure with reasonable accuracy, since the assumption of independent inputs is not violated for the Sobol $g$-function. The primary distinction among the methods lies in their computational efficiency. To assess computational efficiency in a controlled manner, the two-matrix estimator, the Gaussian copula method, and the autoregressive generative method are implemented within a common Monte Carlo framework. The two-matrix estimator generates conditional input samples through column swapping,

**Table 3. Comparison of full total order sensitivity indices for the Sobol g-function across different GSA methods. Estimates are reported as bootstrap means.**

| Index | Sobol | Copula | PCE | AR | Diffusion | Analytical |
|---|---|---|---|---|---|---|
| $S_{T_1}^{\text{full}}$ | 0.27 | 0.27 | 0.22 | 0.30 | 0.28 | 0.28 |
| $S_{T_2}^{\text{full}}$ | 0.24 | 0.23 | 0.24 | 0.26 | 0.23 | 0.24 |
| $S_{T_3}^{\text{full}}$ | 0.22 | 0.25 | 0.26 | 0.24 | 0.22 | 0.21 |
| $S_{T_4}^{\text{full}}$ | 0.20 | 0.17 | 0.17 | 0.20 | 0.17 | 0.19 |
| $S_{T_5}^{\text{full}}$ | 0.15 | 0.15 | 0.13 | 0.16 | 0.15 | 0.16 |
| $S_{T_6}^{\text{full}}$ | 0.13 | 0.13 | 0.14 | 0.14 | 0.14 | 0.15 |
| $S_{T_7}^{\text{full}}$ | 0.11 | 0.11 | 0.13 | 0.11 | 0.12 | 0.13 |
| $S_{T_8}^{\text{full}}$ | 0.11 | 0.10 | 0.12 | 0.09 | 0.12 | 0.12 |
| $S_{T_9}^{\text{full}}$ | 0.08 | 0.09 | 0.11 | 0.10 | 0.11 | 0.10 |
| $S_{T_{10}}^{\text{full}}$ | 0.09 | 0.09 | 0.11 | 0.09 | 0.10 | 0.10 |
| MAE | $1.6 \times 10^{-3}$ | $1.7 \times 10^{-3}$ | $8.9 \times 10^{-3}$ | $1.9 \times 10^{-3}$ | $1.4 \times 10^{-3}$ | 0.00 |

whereas the Gaussian copula and autoregressive models additionally apply transformations based on learned conditional dependencies to generate new input samples after each column swap, following the procedure outlined in Algorithm 1. For these methods, the total number of model evaluations required by the Monte Carlo procedure is $n_{\text{MC}} = N_2 \times (m + 2)$, where $N_2 = N_S$. In contrast, the diffusion based method described in Algorithm 2 requires $n_{\text{MC}} = N_V + N_2 \times (m - 1)$ model evaluations, where $N_2 = M \times N_I \times N_O$. For a fair comparison, we fix $N_V = 10{,}000$, $N_I = 3$, and $N_O = 1$ as the optimal configuration proposed in [7], and vary $N_S$ while selecting $M$ such that all methods use the same total number of model evaluations $n_{\text{MC}}$. The resulting computational efficiency analysis is reported in Fig A in S3 Appendix. It shows that Algorithm 2 reaches stability much faster than Algorithm 1, but exhibits larger uncertainty at convergence due to the heuristic nature of the conditional sample generation method induced by RePaint as well as the unavoidable bias resulting from the limited numbers of $N_I$ and $N_O$ samples.

Finally, we compare the computational running times of the different GSA methods. Each Monte Carlo step requires one model evaluation, whose average computational cost over the input space is denoted by $T_e$. Methods that involve surrogate modeling of the input distribution require an additional set of samples to train the surrogate model, with the associated training cost denoted by $T_t$. These methods also incur an additional cost for generating conditional samples at each Monte Carlo step according to the learned dependency structure; the average cost of this step is denoted by $T_s$. The PCE method is an exception, as it computes the sensitivity indices analytically from the estimated polynomial coefficients. The total computational cost can therefore be expressed as

$$T_{\text{total}} = T_t + n_{\text{MC}} \times (T_s + T_e).$$

To leverage the parallel architecture of the GPU, we employ vectorized batching for both training and sampling. For the autoregressive method, the training cost $T_t$ is optimized by training all $m = 200$ conditional surrogates simultaneously using vectorized ensembling. Furthermore, the sampling costs $T_s$ for both the autoregressive and diffusion methods are reported as effective per-sample costs, achieved by amortizing the iterative sampling process over a batch size of *1024*. The resulting performance comparison is summarized in Table 4.

From Table 4, we can see that the generative model generally requires a longer running time than traditional methods in this high-dimensional problem to achieve similar performance. The increased computational cost arises from the repetitive conditional sample generation process based on the trained generative models. However, this additional

**Table 4. Comparison of running times of different GSA methods for the Sobol _g_-function.**

| Time | Sobol | Copula | PCE | AR | Diffusion |
|------|-------|--------|-----|-----|-----------|
| $T_e$ | 0.08 ms | 0.08 ms | N/A | 0.08 ms | 0.08 ms |
| $T_s$ | N/A | 0.10 ms | N/A | 0.15 ms | 0.24 ms |
| $T_t$ | N/A | 49 s | 4.1 min | 3.5 min | 3.9 min |
| $T_{total}$ | 5.4 min | 6.7 min | 4.1 min | 19 min | 25 min |

All reported running times correspond to GPU wall-clock time measured on an NVIDIA A1000 GPU.

computational expense is justified by the substantially improved accuracy of sensitivity estimates in the presence of complex input correlations, as demonstrated in previous examples, where traditional methods fail to handle such dependencies effectively. The diffusion model based GSA framework can be further adapted by training the diffusion model to learn the joint distribution of inputs and outputs. In this setting, the RePaint algorithm can be used to generate output samples directly, thereby avoiding explicit evaluations of the original model. This modification is particularly beneficial when the cost of evaluating the model output for a given input is high and comparable to $T_s$. By eliminating the need for repeated model evaluations, this approach yields a substantial reduction in the overall running time of Algorithm 2.

## Epidemiology Model

In this section, we present a case study on the epidemiological modeling of the COVID-19 outbreak in Wuhan during the winter of 2019. This example illustrates how a mathematical model calibrated using standard MCMC algorithms can be integrated with an autoregressive generative model to perform GSA. The autoregressive model is chosen in this context because the underlying SIR model involves only a few parameters to estimate and features minimal mathematical complexity, making it particularly well-suited for this approach.

We begin by considering the classical SIR model to analyze the spread of COVID-19 in Wuhan. The data set $D$ consists of the daily number of newly confirmed cases reported by the China CDC between January 21 and February 4, 2020 [38]. Fig 1 displays both the transmission diagram and the corresponding system of differential equations. In this model, the compartment $S$ denotes the susceptible population, $I$ denotes the infected population, and $R$ represents the removed population, which includes both diagnosed cases and recovered individuals. The model includes four parameters: the transmission rate $\beta$, the diagnosis rate $\rho$, the recovery rate $\gamma$, and the initial number of infected individuals $I_0$. These parameters form the vector $\theta = (\beta, \rho, \gamma, I_0) \in \mathbb{R}^4$, which we calibrate using the observed data $D$.

To calibrate the model, we apply the Bayesian approach described in Bayesian Inference section, generating a thinned posterior sample set $S$ of size $N = 10,000$, which provides an empirical approximation of the posterior distribution $p(\theta \mid D)$. From $S$, we qunatify the 95% credible intervals to characterize the plausible ranges of the parameters under the current model and data assumptions. Table A in S4 Appendix summarizes the parameter definitions and their calibrated values. The model output, or QoI, is the total number of individuals infected over the course of the outbreak, defined as

$$\frac{dS}{dt} = -\beta SI,$$
$$\frac{dI}{dt} = \beta SI - \rho I - \gamma I,$$
$$\frac{dR}{dt} = \rho I + \gamma I.$$

**Fig 1. SIR model.** Compartmental transmission diagram and corresponding system of ODEs.

$$y = \int_0^T \beta S(t) I(t) \, dt.$$

To illustrate why traditional GSA methods cannot be applied directly in this setting, we present the posterior predictive distribution of $I(t)$, along with the distributions obtained after swapping the $\beta$ variables between two independent sample sets. The resulting distributions of $I(t)$ based on Sobol conditional samples are shown in Fig 2. We observe that, after swapping, the distribution becomes substantially more dispersed than the original posterior predictive distribution. This inflated variability can lead to sensitivity indices exceeding one, as the underlying correlations among inputs are ignored, thereby inducing unrealistically high volatility in the model outputs. To estimate Sobol sensitivity indices while accounting for correlations among model parameters $\theta$, we use $m = 4$ Rosenblatt transformations, each corresponding to a cyclic permutation of $\theta$. These transformations are approximated by training four autoregressive generative models. Each model learns a Rosenblatt transformation $RT_i$ that maps the posterior distribution $p(\theta \mid D)$ to the uniform distribution over the unit hypercube, following a specific cyclic ordering starting with $\theta_i$.

Since the SIR model has a four-dimensional input, which is comparable to previous benchmark examples, we adopt the same neural network architecture consisting of two hidden layers with 32 neurons each. As discussed in the Ishigami Function example, a training sample size of $N_1 = 6,000$ typically provides good performance for low-dimensional input distributions. We therefore select the later $N_1$ samples from the posterior sample set $S$ to train the autoregressive generative models. To validate the generative models, we compare the empirical marginal distributions estimated from samples generated under different permutation orders with those estimated from the posterior sample set $S$. These comparisons, shown in Fig 3, demonstrate strong agreement for parameters with approximately Gaussian marginal posterior densities, indicating that the models reliably reproduce the target distribution. For parameters with heavy-tailed marginal densities, such as $I_0$ shown in Fig 3C, the generated samples exhibit bias near the tail. Nevertheless, the model performs acceptably well overall. These comparisons confirm that the generative model accurately captures the posterior distributions, which is essential for reliable sensitivity indices estimation.

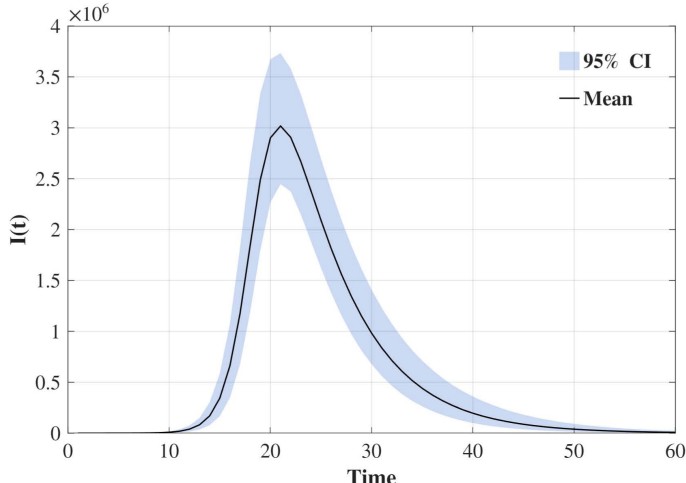

**A.** Posterior predictive distribution of $I(t)$

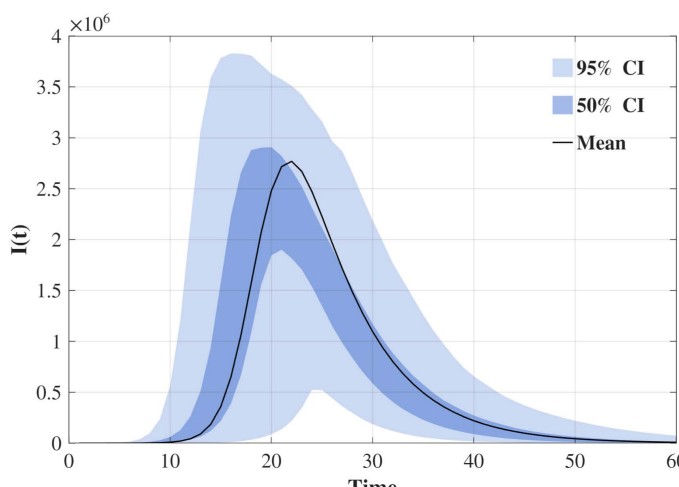

**B.** Sobol conditional samples $I(t)$

**Fig 2. Failure of Sobol's method.** Column swapping in Sobol's two-matrix estimator ignores Bayesian-calibrated parameter correlations, leading to distorted predictive distributions of $I(t)$.

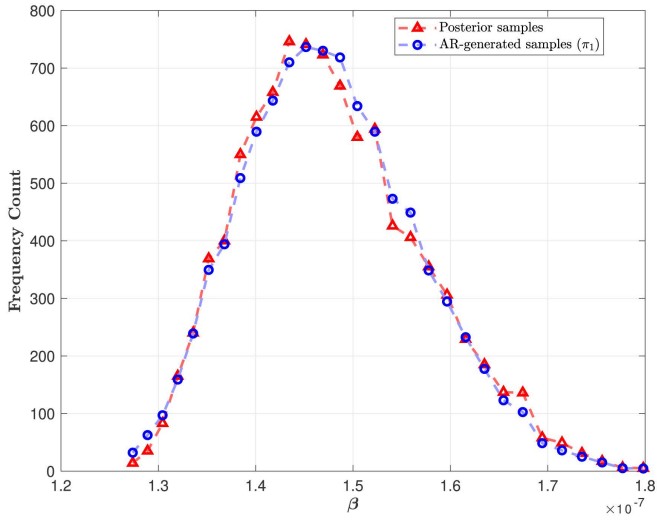

**A.** Empirical marginal density of $\beta$

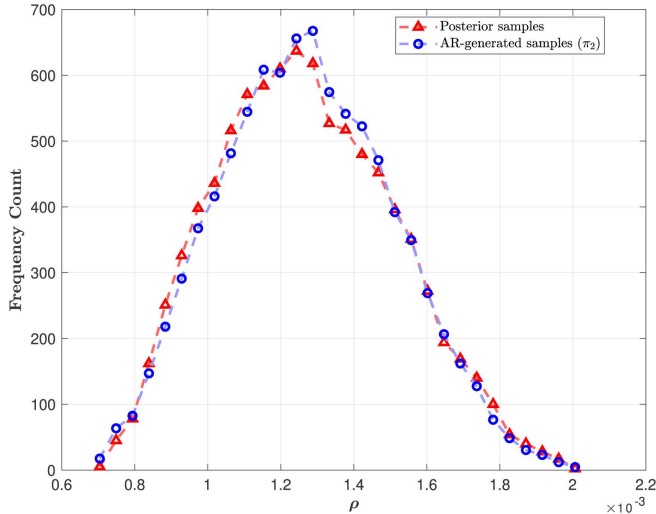

**B.** Empirical marginal density of $\rho$

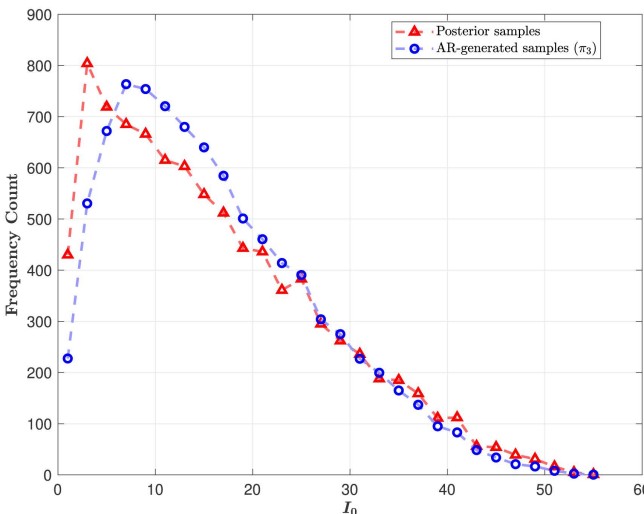

**C.** Empirical marginal density of $I_0$

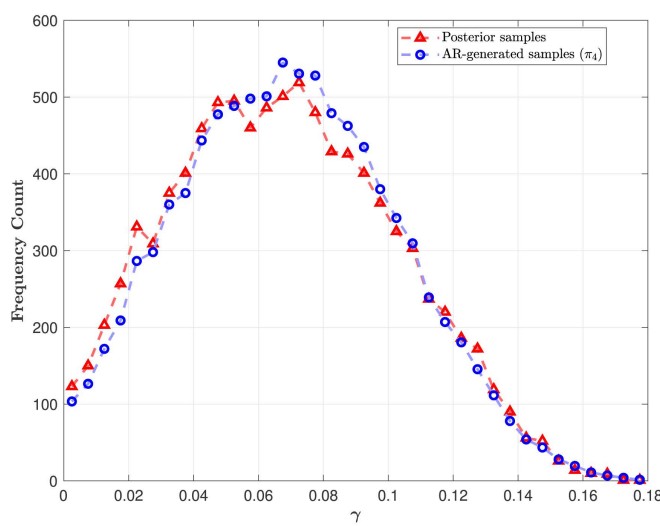

**D.** Empirical marginal density of $\gamma$

**Fig 3. Comparison.** Empirical marginal density comparison between posterior samples and autoregressive model generated samples.

With the trained autoregressive models, we apply Algorithm 1, setting the Monte Carlo sample size to $N_2 = 2,000$ and using 2,000 bootstrap replicates, to compute the full Sobol's indices $S_i^{\text{full}}$, $S_{T_i}^{\text{full}}$, as well as the independent indices $S_i^{\text{ind}}$ and $S_{T_i}^{\text{ind}}$, for all calibrated model parameters within the SIR model shown in Fig 1. A summary of the results is presented in Fig 4. Based on the estimated Sobol sensitivity indices, we conclude that the transmission rate $\beta$ is overall the most influential parameter in determining the total number of infected individuals, followed by the case detection rate $\rho$. The

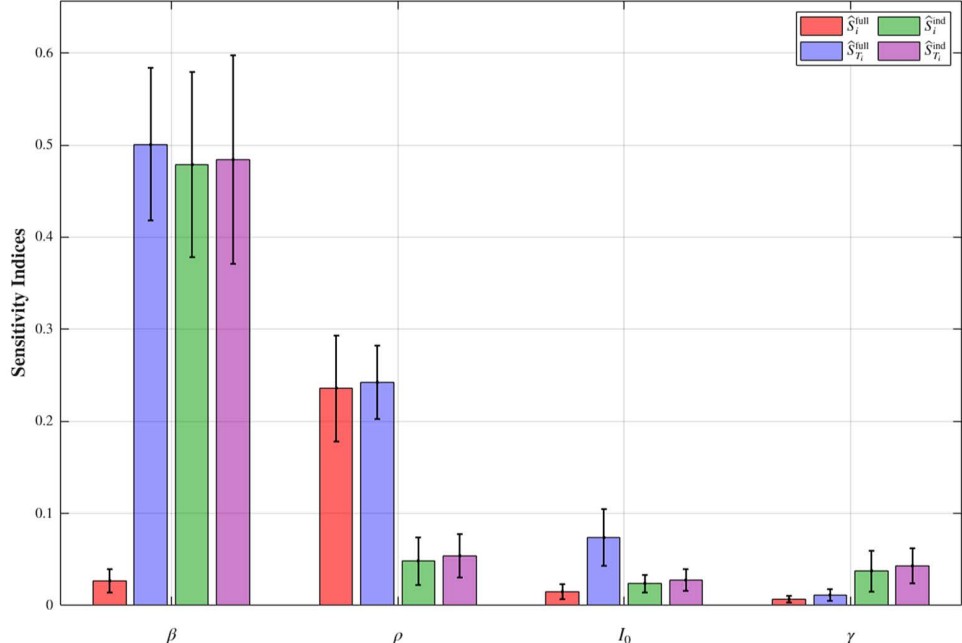

**Fig 4. Results.** GSA results for the SIR model using an autoregressive model. Estimates are reported as bootstrap means, with error bars indicating 95% bootstrap confidence intervals.

remaining parameters, including the initial number of infectious individuals $I_0$ and the recovery rate $\gamma$, are relatively less important. Some parameters, especially $\beta$, have their full total-order sensitivity index values significantly larger than their full first-order sensitivity index values. This indicates that when considering each parameter itself as well as its correlations with others, the majority of the sensitivity comes from the interaction between $\beta$ and its correlations with other parameters. This pattern is similarly true for $I_0$, though it is treated as insensitive to model output. The $\rho$ parameter has all its full sensitivity indices greater than its independent sensitivity indices, which means that when considering the parameter itself independently, $\rho$ is treated as an important factor. However, when considering the full correlation structure with other parameters, $\rho$ is not as sensitive compared to $\beta$.

From a practical decision-making perspective, researchers commonly adopt a threshold of 0.1 for Sobol sensitivity indices [16]; parameters with at least one sensitivity index exceeding this threshold are typically considered sensitive, as they explain a non-negligible portion of the variance in model outputs. In the context of the SIR model applied to the COVID-19 epidemic, the parameters $\beta$ and $\rho$ are identified as influential. This finding suggests that public health interventions aimed at reducing the transmission rate, such as implementing quarantines and lockdowns, are among the most effective strategies for controlling the size of the infected population. Furthermore, the sensitivity of $\rho$ highlights the importance of expanding testing capacity to increase the detection rate, which in turn helps reduce the final epidemic size.

## Cellular Dynamic Model

CAR-T cell therapy represents a groundbreaking immunotherapy for treating resistant hematologic cancers by utilizing genetically modified T cells to target tumor specific antigens. Despite its clinical success, the therapy faces challenges such as cytokine release syndrome (CRS) and CAR-T cell exhaustion, which limit its safety and long-term efficacy. A deep understanding of the complex dynamics of CAR-T cell proliferation, differentiation, and tumor interaction is essential for optimizing treatment outcomes and advancing personalized medicine. Mathematical modeling, when calibrated with

patient-specific therapy data, offers a powerful framework to capture these dynamics, predict therapeutic responses, and inform clinical decision-making [39].

In this experiment, we consider a dynamical systems model that describes the interactions between CAR-T cell phenotypes and tumor cells. The model is illustrated through a transition diagram and a system of ODEs shown in Fig 5.

In this model, four CAR-T cell subpopulations are considered: distributed ($C_D$), effector ($C_T$), memory ($C_M$), and exhausted ($C_E$), along with tumor cells ($T_u$). The system accounts for transitions between these subpopulations, tumor growth, and cell death due to natural processes and tumor-killing mechanisms. The transition diagram illustrates the interactions, with corresponding differential equations governing the population dynamics. The functional CAR-T cell population, responsible for tumor killing, is defined as $C_F = C_D + C_T$, which includes both distributed and effector cells. The effector expansion rate is modeled as a time-dependent function,

$$k(t) = r_{\min} + \frac{p_1}{1 + (p_2 t)^{p_3}},$$

which captures the transient proliferation phase following cell infusion. Antigen-stimulated proliferation is governed by

$$F(T_u) = \frac{T_u}{A + T_u},$$

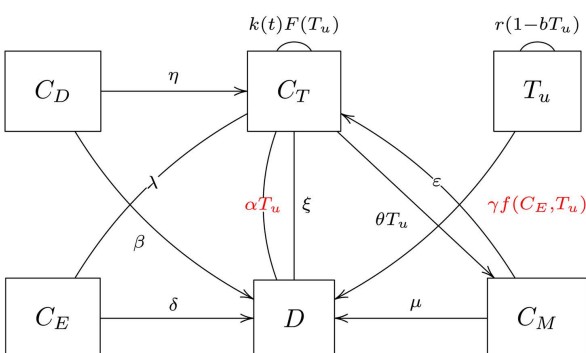

$$\frac{dC_D}{dt} = -(\beta + \eta)C_D,$$

$$\frac{dC_T}{dt} = \eta C_D + k(t)F(T_u)C_T - (\xi + \varepsilon + \lambda)C_T + \theta T_u C_M - \alpha T_u C_T,$$

$$\frac{dC_M}{dt} = \varepsilon C_T - \theta T_u C_M - \mu C_M,$$

$$\frac{dC_E}{dt} = \lambda C_T - \delta C_E,$$

$$\frac{dT_u}{dt} = r T_u (1 - b T_u) - \gamma f(C_E, T_u) T_u.$$

**Fig 5. CAR-T model.** Compartmental transmission diagram and corresponding system of ODEs.

reflecting saturation kinetics with respect to tumor burden. Tumor cytotoxicity is described by the nonlinear function

$$f(C_F, T_u) = \frac{\frac{C_F}{T_u}}{v + \frac{\alpha + C_F}{T_u}},$$

which accounts for complex interactions between CAR-T cells and tumor cells, including immune escape mechanisms. To calibrate the model, we fit it to clinical measurements of total CAR-T cell counts, modeled as $C = C_D + C_T + C_M + C_E$, using data from a single patient recorded over a one-month period post-infusion [40]. The calibration process is performed using a Bayesian MCMC algorithm, generating $N = 50{,}000$ thinned posterior samples to approximate the posterior distribution of the model parameters. Table A in S5 Appendix summarizes the detailed definitions of all model parameters along with their calibrated values.

Two clinically significant quantities can be defined from the model dynamics. The peak functional CAR-T cell count, $C_F^{max} = \max_t(C_D(t) + C_T(t))$, represents the maximum cytotoxic potential of the therapy and is strongly associated with both tumor clearance and the risk of CRS. The exhausted CAR-T cell fraction, defined as $C_{E_p} = C_E(T)/C(T)$ quantifies the proportion of exhausted CAR-T cells relative to the total CAR-T cell population at the end of the observation period. A high exhausted fraction indicates reduced CAR-T cell functionality, correlating with treatment failure or relapse, and is a critical marker for assessing therapy durability. In order to analyze which factor is most impoartant the the two quiaties, GSA can be applied. Since the model contains $m = 13$ kinetic parameters which we calibrated from the data observations, we let the parameter vector $\theta \in \mathbb{R}^{13}$ include all the kinetic parameters. Since the model has a complex structure and the parameters lie in a relatively high-dimensional space, we apply the proposed GSA method using the diffusion model to estimate the Shapley effect based sensitivity indices. Again, we employ a standard two layer MLP with 32 neurons per layer as the noise prediction network in the diffusion model. The later $N_1 = 15{,}000$ samples from the posterior sample set $S$ were used to train the diffusion model. In S5 Appendix, we illustrate the approximation quality and the learned posterior distribution error, demonstrating that this sample size is sufficient while avoiding excessive computational cost. With the trained diffusion model, we set the number of permutations to $M = 1 \times 10^5$ to ensure convergence of Algorithm 2. The resulting GSA estimates, together with bootstrap confidence intervals, are summarized in Fig 6.

Based on Fig 6A, we identify four parameters as dominant contributors to the variability in $C_F^{max}$: the transition rate from distributed to effector cells $\eta$, the infusion rate of CAR-T cells $\beta$, the exhaustion rate $\lambda$, and the baseline rate $r_{min}$ of the time-dependent proliferation rate function $k(t)$. Each parameter exhibits at least one sensitivity index exceeding the 0.1 threshold, indicating substantial influence on the peak functional CAR-T cell count. This finding aligns with the biological interpretation of $C_F^{max}$ as reflecting the early expansion and differentiation dynamics of CAR-T cells.

The parameters $\beta$ and $\eta$ control the activation of distributed cells into the effector population, directly increasing the pool of cytotoxic CAR-T cells. The exhaustion parameter $\lambda$ governs the transition from effector to exhausted cells, modulating the functional lifespan of $C_T$ and thereby limiting the duration of effector cell contribution to tumor clearance. The proliferation function $k(t)$, which captures the transient expansion of effector cells post-infusion, is determined by four parameters: the baseline rate $r_{min}$ and Hill function parameters $p_1$, $p_2$, and $p_3$. Among these, $r_{min}$ is the most influential, while the combined effect of all four parameters accounts for approximately 16.7% of the total variance in $C_F^{max}$, as assessed by summing their Shapley sensitivity indices. This aggregation underscores a key advantage of Shapley effects: their additive property enables straightforward group-wise interpretation of parameter influence. This is particularly useful when parameters collectively define a functional component of the model, such as $k(t)$. Other parameters exhibit relatively low sensitivity. In particular, the weak influence of $\gamma$ suggests that, although tumor burden can affect CAR-T cell expansion through antigen stimulation, the peak cytotoxic potential is mainly determined by the internal expansion and transmission of CAR-T cells.

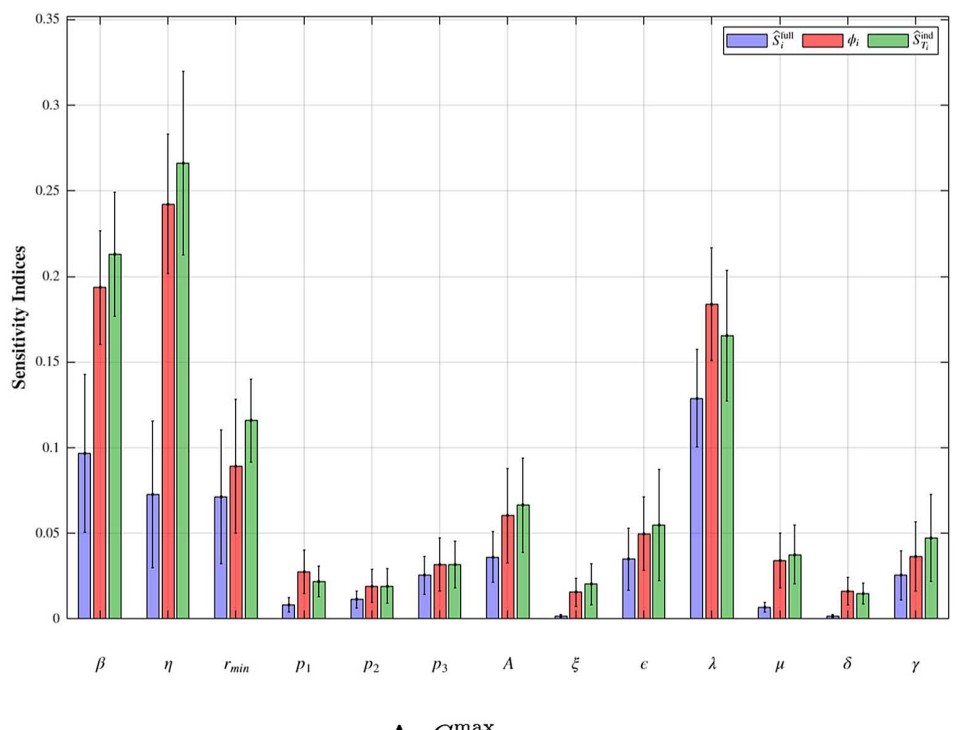

**A.** $C_F^{\max}$

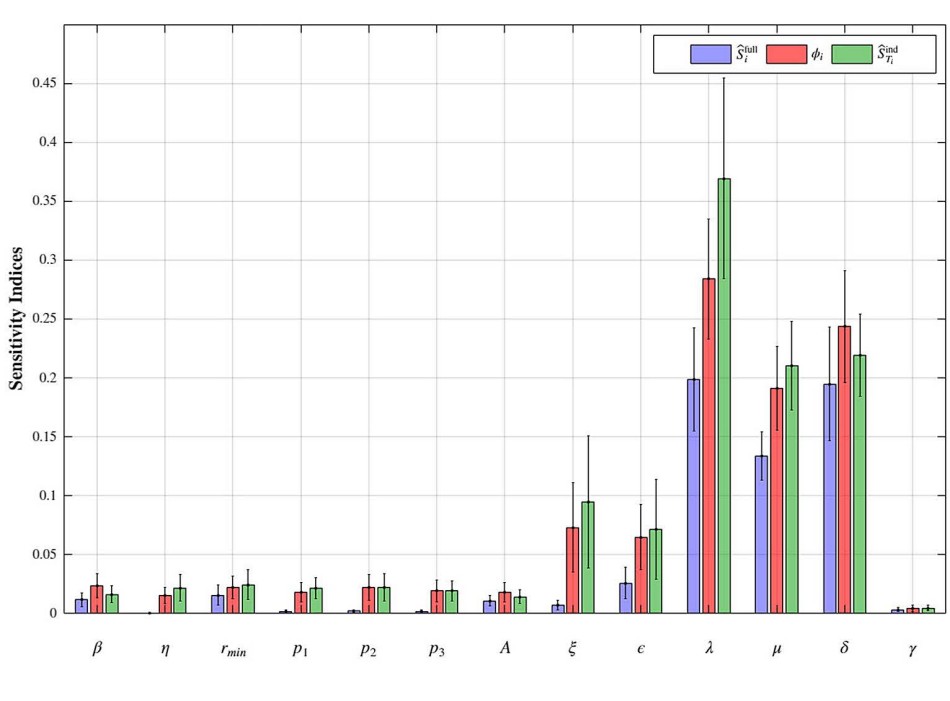

**B.** $C_{E_p}$

**Fig 6. Results.** GSA results for the CAR-T model using a diffusion model. Estimates are reported as bootstrap means, with error bars indicating 95% bootstrap confidence intervals.

Fig 6B shows a significantly different sensitivity andscape for the exhausted CAR-T cell fraction $C_{E_p}$. Here, the most influential parameters are $\lambda$, $\mu$, and $\delta$. Specifically, $\lambda$ again plays a central role by directly governing the flow of effector cells into the exhausted compartment. The memory cell clearance rate $\mu$ indirectly impacts exhaustion by regulating the reservoir of $C_M$, which may re-enter the effector cell population. The exhausted cell clearance rate $\delta$ influences the steady-state level of $C_E$ at the end of the observation window, thereby affecting the exhaustion ratio. Notably, parameters that strongly affect early proliferation, such as $\eta$ and $k(t)$ related parameters, are far less impactful here, indicating a clear distinction between expansion and long-term exhaustion processes.

These findings provide actionable insights for optimizing CAR-T therapy. To enhance peak functional response, characterized by a higher value of $C_F^{\max}$, strategies should prioritize promoting early effector expansion while delaying premature exhaustion. This could include optimizing infusion protocols to increase $\eta$, or pharmacologically modulating pathways related to $r_{\min}$ and $\lambda$. To improve long-term therapy durability, as indicated by a lower exhausted CAR-T cell fraction $C_{E_p}$, interventions may target both limiting the exhaustion rate and increasing the proportion and longevity of memory CAR-T cells. Furthermore, sensitivity analysis offers a systematic way to identify which parameters require more precise patient-specific estimation during model calibration and which can be fixed or marginalized, thus streamlining clinical model personalization.

## Discussion

In this study, we present methods that integrate data-informed mathematical models with generative AI to address the challenge of generating accurate conditional samples for estimating variance-based global sensitivity indices. We introduce two complementary approaches: an autoregressive model with Rosenblatt transformations and a diffusion model with Shapley effects, both designed to learn parameter dependencies directly from Bayesian posterior samples.

The autoregressive model enables numerically efficient two-matrix estimators for Sobol's indices by training separate models on posterior samples with different cyclic permutations, making it particularly suitable for low-dimensional models with computationally inexpensive simulations. In contrast, the diffusion model learns parameter dependencies through a single training phase and can generate parameter samples conditioned on arbitrary parameter subsets without requiring additional training. This design is compatible with numerically efficient random Shapley estimators and ensures scalability for high-dimensional problems.

We evaluated our methods across problems of varying complexity. A linear Gaussian example validated numerical accuracy of our methods against analytical solutions. The COVID-19 model illustrated the practical utility of the autoregressive method in epidemiological modeling, demonstrating its broad applicability for public health applications. The CAR-T cell dynamics model demonstrated the feasibility of applying diffusion based GSA to highly complex mathematical systems, making it well-suited for problems involving intricate biological processes, such as those in systems biology and neuroscience. When dealing with extremely high-dimensional parameter spaces, replacing MLPs with U-Net architectures in the diffusion model can enhance performance. The availability of open-source software packages facilitates the adoption of generative AI based global sensitivity analysis methods across a wide range of scientific disciplines.

Several limitations merit consideration. The quality of sensitivity estimates depends on the accuracy of posterior samples; poor MCMC convergence, especially in high-dimensional spaces, can introduce bias into the estiamted sensitivity indices. The current autoregressive implementation assumes Gaussian conditional distributions, which may be inadequate for non-Gaussian scenarios characterized by heavy tails or multi-modality. This limitation underscores the need for more expressive yet invertible alternatives for modeling conditional dependencies. In the case of diffusion models, the Repaint algorithm used for conditional sampling has been demonstrated to be a heuristic approach. While it produces samples that match the joint distribution characteristics, it does not guarantee exact sampling from the actual conditional distributions. Although exact conditional samplers, such as sequential Monte Carlo methods, have been proposed [41], they are often computationally prohibitive for practical use. Given these limitations, future improvements could involve incorporating

advanced posterior inference methods such as nested sampling or variational approximations to improve robustness in high-dimensions. Additionally, further development of the autoregressive architecture to accommodate non-Gaussian conditionals and integrating exact conditional samplers into diffusion models may enhance performance and broaden applicability across complex model classes.

Beyond deterministic models, stochastic models such as Stochastic Differential Equations (SDEs) represent another important class of mathematical models. These models introduce randomness into the system dynamics, providing more realistic representations in certain contexts such as individual contact networks in infectious disease related research. The proposed framework holds potential for advancing GSA for stochastic models. Methods like DDPMs can be adapted to learn data-calibrated joint posterior distributions over outputs and parameters, thereby capturing the randomness inherent in stochastic simulations. Conditional sample generation can then be employed to produce model outputs directly, eliminating the need for repeated simulations when estimating sensitivity indices. An enhanced methodology could be developed to enable effective GSA across a broader class of data-informed mathematical models, including stochastic systems, by integrating approximate Bayesian calibration with generative models. This approach paves the way for future research.

## Supporting information

**S1 Appendix. Linear Function.** Supplementary results comparing the true input distribution with the generative model–learned distribution, together with detailed GSA results and bootstrap confidence intervals.
(PDF)

**S2 Appendix. Ishigami Function.** Supplementary results comparing the true input and conditional distributions with the learned generative distributions, including effective sample size diagnostics and detailed GSA results with bootstrap confidence intervals.
(PDF)

**S3 Appendix. Sobol Function.** Supplementary results assessing the efficiency of generative model based GSA, together with detailed bootstrap GSA estimates. This appendix also includes conditional density estimation results for autoregressive and diffusion models.
(PDF)

**S4 Appendix. Epidemiological Model.** Supplementary results providing detailed parameter definitions and Bayesian calibration results for the SIR model.
(PDF)

**S5 Appendix. Cellular Dynamics Model.** Supplementary results presenting detailed parameter definitions and Bayesian calibration results, as well as effective sample size diagnostics for posterior samples used to train the diffusion model.
(PDF)

## Author contributions

**Conceptualization:** Xuyuan Wang.

**Data curation:** Xuyuan Wang.

**Formal analysis:** Xuyuan Wang.

**Funding acquisition:** Xuyuan Wang.

**Investigation:** Xuyuan Wang.

**Methodology:** Xuyuan Wang.

**Project administration:** Xuyuan Wang.

**Resources:** Xuyuan Wang.

**Software:** Xuyuan Wang.

**Validation:** Xuyuan Wang.

**Visualization:** Xuyuan Wang.

**Writing – original draft:** Xuyuan Wang.

**Writing – review & editing:** Xuyuan Wang.

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
