## [Decision Letter · Decision Letter 0]

19 Nov 2025

Data-informed global sensitivity analysis for mathematical models using generative AI

PLOS Computational Biology

Dear Dr. Wang,

Thank you for submitting your manuscript to PLOS Computational Biology. After careful consideration, we feel that it has merit but does not fully meet PLOS Computational Biology's publication criteria as it currently stands. Therefore, we invite you to submit a revised version of the manuscript that addresses the points raised during the review process.

We look forward to receiving your revised manuscript.

Kind regards,

Katharina Kusejko

Academic Editor

PLOS Computational Biology

Roger Kouyos

Section Editor

PLOS Computational Biology

**Journal Requirements:**

3) We notice that your supplementary information is included in the manuscript file. Please remove them and upload them with the file type 'Supporting Information'. Please ensure that each Supporting Information file has a legend listed in the manuscript after the references list.

**Reviewers' comments:**

Reviewer's Responses to Questions

**Comments to the Authors:**

Reviewer #1: This manuscript presents a data-driven approach for global sensitivity analysis (GSA) using generative machine learning models. It addresses the challenge of performing GSA on parameters with strong and potentially high-dimensional correlations by leveraging generative models to sample from complex distributions associated with the formulations of Rosenblatt transformations and Shapley effects. Three illustrative examples are provided to demonstrate the proposed framework.

Overall, the manuscript is well written and logically organized. However, several important points require clarification before it can be considered for publication:

(1) The novelty of the proposed method should be more clearly articulated. Beyond the direct integration of generative models into the formulations of Rosenblatt transformations and Shapley effects, are there any methodological innovations or theoretical insights that distinguish this work?

(2) The generative models are trained using MCMC samples, which are inherently dependent. The authors are recommended to discuss how this dependency affects model training and performance, such as in terms of convergence behavior, sampling efficiency, and the reliability of the learned distributions.

(3) A key concern is the limited quantitative validation of the proposed method. Currently, only a low-dimensional linear model (three variables) is used for quantitative assessment. Given that high-dimensional settings are a primary motivation for this work, it is strongly recommended to include quantitative validations for higher-dimensional problems to substantiate the applicability and robustness.

(4) Another major concern is about the sample size required to make the proposed method effective. In the demonstrative cases, the samples used to train the generative models are actually pretty large. Specifically, 50,000 samples, 100,000 samples, and 1,000,000 samples for the first, second, and third cases, respectively. However, the dimension of the involved parameters is not too high (no more than 20). It is important to examine how the performance of the proposed method scales with respect to sample size. A sensitivity analysis on the number of training samples would help clarify the practical feasibility of this approach.

Reviewer #2: Please see the attached file for my comments.

**Have the authors made all data and (if applicable) computational code underlying the findings in their manuscript fully available?**

Reviewer #1: Yes

Reviewer #2: Yes

PLOS authors have the option to publish the peer review history of their article (what does this mean? ). If published, this will include your full peer review and any attached files.

**Do you want your identity to be public for this peer review?** For information about this choice, including consent withdrawal, please see our Privacy Policy .

Reviewer #1: No

Reviewer #2: No

**Figure resubmission:**

**Reproducibility:**



---

## [Decision Letter · Decision Letter 1]

3 Feb 2026

PCOMPBIOL-D-25-01368R1

Bayesian-calibrated global sensitivity analysis for mathematical models using generative AI

PLOS Computational Biology

Dear Dr. Wang,

Thank you for submitting your manuscript to PLOS Computational Biology. After careful consideration, we feel that it has merit but does not fully meet PLOS Computational Biology's publication criteria as it currently stands. Therefore, we invite you to submit a revised version of the manuscript that addresses the points raised during the review process.

We look forward to receiving your revised manuscript.

Kind regards,

Katharina Kusejko

Academic Editor

PLOS Computational Biology

Roger Kouyos

Section Editor

PLOS Computational Biology

**Journal Requirements:**

1) Please upload a copy of Figures 1, and 5 which you refer to in your text on pages 20, and 22. Or, if the figure is no longer to be included as part of the submission please remove all reference to it within the text.

**Reviewers' comments:**

Reviewer's Responses to Questions

**Comments to the Authors:**

Reviewer #1: The reviewer would like to thank the author for the impressive, solid revision. All the concerns of the reviewer have been adequately addressed. The reviewer now strongly recommends this manuscript for publication with great confidence.

Reviewer #2: Thanks for revising the manuscript. Your revision has improved the manuscript significantly and I am happy to recommend for acceptance after you have improved the quality of Figure 2 in the manuscript. The fonts and ticks are too small.

**Have the authors made all data and (if applicable) computational code underlying the findings in their manuscript fully available?**

Reviewer #1: Yes

Reviewer #2: Yes

PLOS authors have the option to publish the peer review history of their article (what does this mean? ). If published, this will include your full peer review and any attached files.

**Do you want your identity to be public for this peer review?** For information about this choice, including consent withdrawal, please see our Privacy Policy .

Reviewer #1: **Yes:** Ziluo Xiong

Reviewer #2: No

**Figure resubmission:**
---

## [Editor Report · Decision Letter 2]

10 Feb 2026

Dear Mr. Wang,

We are pleased to inform you that your manuscript 'Bayesian-calibrated global sensitivity analysis for mathematical models using generative AI' has been provisionally accepted for publication in PLOS Computational Biology.

Best regards,

Katharina Kusejko

Academic Editor

PLOS Computational Biology

Roger Kouyos

Section Editor

PLOS Computational Biology

---

## [Editor Report · Acceptance letter]

PCOMPBIOL-D-25-01368R2

Bayesian-calibrated global sensitivity analysis for mathematical models using generative AI

Dear Dr Wang,

I am pleased to inform you that your manuscript has been formally accepted for publication in PLOS Computational Biology. Your manuscript is now with our production department and you will be notified of the publication date in due course.

With kind regards,

Anita Estes
